# *Plasmodium* ARK1 regulates spindle formation during atypical mitosis and forms a divergent chromosomal passenger complex

Annu Nagar[1,9], Ryuji Yanase [2,9], Mohammad Zeeshan[2,8,9], David J. P. Ferguson [3], Steven Abel[4], Sarah L. Pashley[2], Akancha Mishra[2], Anthonius Eze [2], Edward Rea[2], Declan Brady[2], Andrew R. Bottrill [5], Sue Vaughan [3], Karine G. Le Roch [4], David S. Guttery[2], Anthony A. Holder [6], Eelco C. Tromer [7,10] ✉, Pushkar Sharma [1,10] ✉ & Rita Tewari [2,10] ✉

Mitosis in *Plasmodium* spp., the causative agent of malaria, is fundamentally different from model eukaryotes, proceeding via a bipartite microtubule organising centre (MTOC) and lacking canonical regulators such as Polo kinases. During schizogony, asynchronous nuclear replication produces a multinucleate schizont, while rapid male gametogony generates an octaploid nucleus before gamete formation. Here, we identify Aurora-related kinase 1 (ARK1) as a key component of inner MTOC and spindle formation, controlling kinetochore dynamics and driving mitotic progression. Conditional ARK1 depletion disrupts spindle biogenesis, kinetochore segregation, karyokinesis and cytokinesis in both stages, and affects parasite transmission. Interactome analysis shows that ARK1 forms the catalytic core of a non-canonical chromosomal passenger complex (CPC) containing two highly divergent inner centromere proteins (INCENPs), which we term INCENP-A and INCENP-B, and lacking the canonical chromatin-targeting subunits Survivin and Borealin. Comparative genomics suggests that apicomplexan INCENPs arose through recurrent lineage-specific duplications, reflecting an evolutionary rewiring of CPC architecture in this eukaryotic lineage. Together, these findings reveal key adaptations in *Plasmodium* mitosis involving ARK1 and its INCENP scaffolds, and identify the ARK1–INCENP interface as a potential multistage target for antimalarial intervention.

*Plasmodium* spp., the causative agent of malaria, is a haploid unicellular eukaryote of the phylum Apicomplexa, which is evolutionarily distant from most model organisms and has some unusual divergent features of cell division. During its life cycle, mitosis is closed and occurs during two distinct forms of cell division: schizogony within vertebrate host cells, and male gametogony within the female mosquito gut (Fig. 1a). During asexual proliferation within host red blood cells (schizogony), repeated asynchronous nuclear division, without chromosome condensation, produces a multinucleate schizont. Subsequent budding produces invasive merozoites to continue the cycle.

In contrast, during male gametogony three rounds of rapid genome replication to octaploid (1 N to 8 N) occur in 8 min within the intact nucleus. Karyokinesis and cytokinesis are synchronised with axoneme dynamics by replication and separation of a bipartite microtubule-organising centre (MTOC), chromosome condensation, and complementary de novo basal body formation, leading to the budding of eight flagellated male gametes. These events occur within 12–15 min of gametocyte activation, which is probably the most rapid mitosis amongst eukaryotes[1]. Compared to schizogony, mitosis in male gametogony is unusual, with atypical continuous spindles and a bipartite MTOC that spans the nuclear envelope and coordinates the simultaneous assembly of spindle microtubules and the flagellar axoneme. Male gametes are the only flagellated stage in the parasite's life cycle, and therefore the only stage where centriole formation and de novo replication of the basal body occur[1–4].

Aurora-related kinases (ARKs) are multi-functional mitotic regulators that coordinate spindle assembly, chromosome segregation, and late mitotic events across eukaryotes. They act on spindle pole duplication, spindle assembly, kinetochore-microtubule attachment, mitotic checkpoints, and finally cytokinesis[5–8]. Budding yeast and the amoeba *Dictyostelium discoideum*[9] have a single ARK to execute all functions, suggesting this state to be ancestral to all eukaryotes[7]. However, multiple different lineages (e.g., vertebrates, plants, kineto-plastids, and others) have one or more paralogues that resulted from parallel duplications, which subdivided functions over the different paralogues[7,10]. One paralogue (Aurora B in humans) is the catalytic

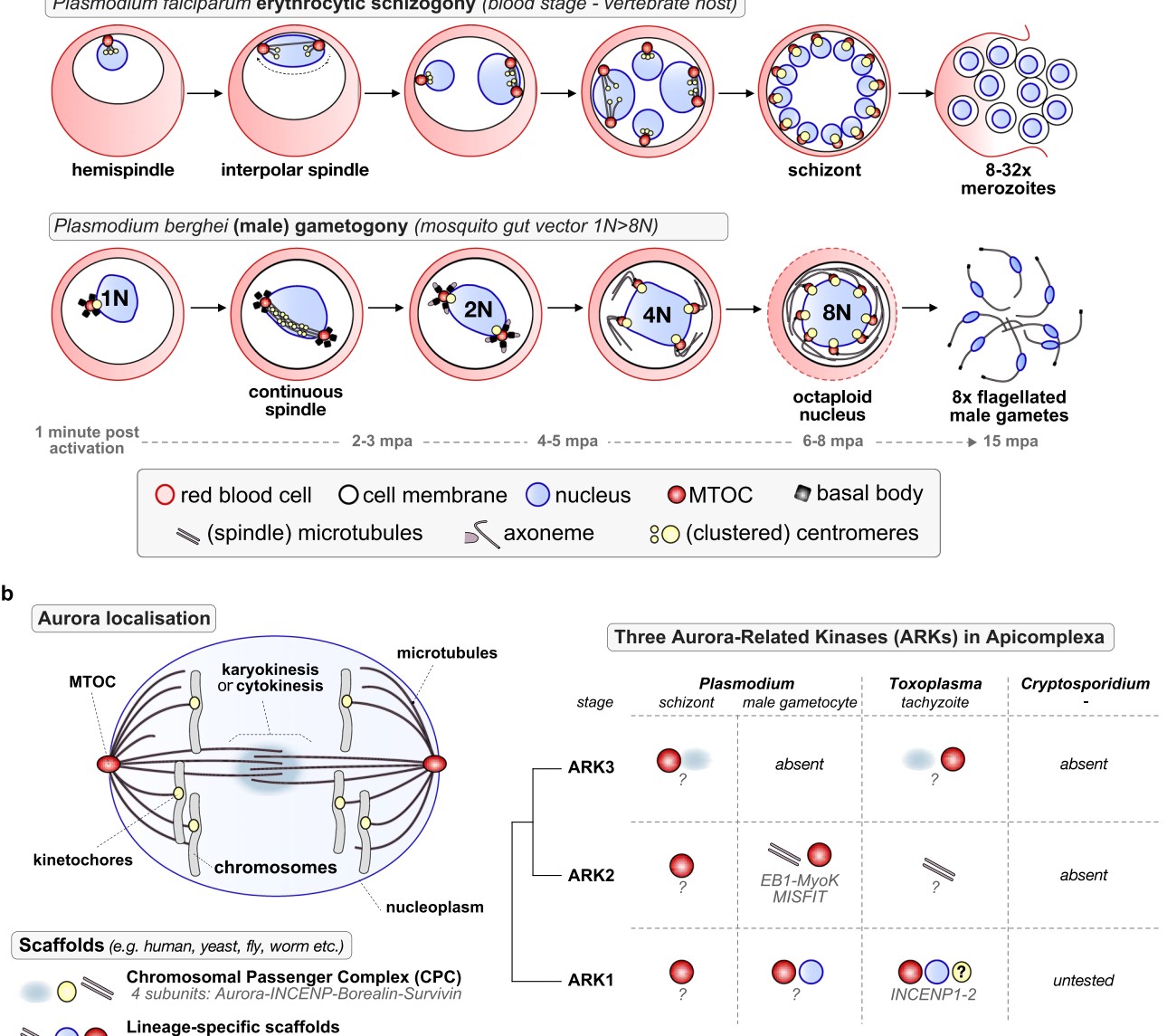

**Fig. 1 | Expansion of the Aurora kinase family among apicomplexan parasites coincides with divergent mitotic mechanisms. a** The schematic shows schizogony in the asexual blood stage and male gametogony in *Plasmodium*, highlighting some of the subcellular structures involved (below). For schizogony: based on U-ExM classifications[23]: sequence of intranuclear MT structures: one hemispindle → bipolar mitotic spindle → interpolar microtubules (the outer plaque is closely associated with the IMC/plasma membrane)[24]. **b** Left: five different cellular locations of Aurora kinases during mitosis/meiosis[10], based on findings from model organisms—projected on the nucleus of *Plasmodium* spp. with a closed nucleus during mitosis and nuclear envelope-embedded MTOCs. The cytokinesis-associated region is shown only as a functional analogue based on ARK3 localisation in *Toxoplasma*[33], and does not imply a canonical central spindle or midbody in *Plasmodium*. Right: Overview of the location (pictograms) and putative protein complex/interactors (italicised text) of three Aurora kinase paralogues ARK1-2-3 in the three apicomplexan parasite models *Plasmodium*, *Toxoplasma* and *Cryptosporidium*.

centre of the chromosomal passenger complex (CPC), a key regulator of mitotic events at the cell equator, which initially hitchhikes on the inner centromere and finally remains at the central spindle to orchestrate cytokinesis. In canonical model systems, the CPC consists of Aurora B, inner centromere protein (INCENP), Survivin, and Borealin[11,12] that guide its localisation to kinetochores, spindle and the central spindle. It is also critical for mitotic histone phosphorylation, fixing errors in kinetochore-spindle attachment, and correct cytokinesis (Fig. 1b).

Apicomplexans, including *Plasmodium* and *Toxoplasma* encode three ARKs, most of which lack canonical localisation and activation motifs found in model eukaryotes[10,13,14]. ARK1, 2, and 3 are refractory to gene disruption and indispensable during asexual blood stage schizogony in both *P. berghei* and *P. falciparum*[13–15]. Consistent with this, a recent study has chemically validated Aurora kinases as promising targets for the development of multi-stage antimalarials[16,17]. ARK1 has been observed at the MTOC during schizogony and in *P. falciparum* gametocytes[18,19]. However, its precise function in asexual division and its behaviour during sexual development have remained unclear. Our recent studies on ARK2 revealed its association with spindle microtubules (MTs) forming a divergent protein complex with EB1, myosin K and MISFIT[10], but little is known of ARK1 and 3 functions during male gametogony or whether they are essential for parasite transmission. ARK1 function has also been explored in another apicomplexan, *Toxoplasma gondii*, where it interacts with two INCENP paralogues and is important for asexual replication. Whether *Plasmodium* INCENPs are direct orthologues of *Toxoplasma* INCENPs is not clear. The function of *Toxoplasma* ARK1 during male gametogony is difficult to study because the sexual cycle of these parasites only takes place in the gut of cats (Fig. 1b)[20].

Here, we have investigated *Plasmodium* ARK1, integrating studies of location, function and molecular interactions across two life-cycle stages using *P. falciparum* (Pf, for asexual blood-stage schizogony) and *P. berghei* (Pb, for male gametogony). High-resolution live-cell imaging and expansion microscopy revealed ARK1 associated with the mitotic spindle, kinetochores, and the inner MTOC. Conditional gene knockout using rapamycin-DiCre in *P. falciparum*[21] and gene knockdown using a promoter-trap approach in *P. berghei*[22] revealed that ARK1 regulates spindle biogenesis, karyokinesis and cytokinesis during both schizogony and male gametogony, suggesting that ARK1 is a critical regulator of *Plasmodium* mitosis. Interactome analysis identifies ARK1 as the catalytic component of a non-canonical Chromosomal Passenger Complex (CPC) comprising two divergent INCENP paralogues–here termed INCENP-A and INCENP-B–in the absence of Survivin and Borealin. Together, these findings reveal how ARK1 and its INCENP scaffolds contribute to the distinctive mechanisms of apicomplexan nuclear division.

## Results

### ARK1 is present in asexual stage parasites undergoing mitotic division

We studied the location of PfARK1 in asexual blood stages, using a rapamycin-inducible dimerizable Cre recombinase DiCre-based strategy to tag and knockout *PfARK1* in the 1G5DC parasite line (Supplementary Fig. 1a)[21]. These parasites (HA-PfARK1-loxP) were used to locate PfARK1 during parasite division using an antibody to the HA-tag included at its N-terminal. Indirect immunofluorescence (IFA) and Western blot indicated that ARK1 is expressed only in trophozoite and schizont stages during asexual division, but was not detectable in ring and segmentor (Fig. 2a, Supplementary Fig. 1e). ARK1 was present as a punctum proximal to the MTOC, as identified by staining with an anti-centrin antibody, present only in cells with two associated centrin puncta, indicative of ongoing mitosis (Fig. 2a, c). These observations were largely consistent with previous studies of PfARK1[18,19].

To gain a deeper insight into ARK1 dynamic organisation, Ultra-structure Expansion Microscopy (U-ExM) was performed, and visualized ARK1 along with spindle microtubules using an anti-tubulin antibody to track spindle biogenesis. As reported previously[23], parasites possessed hemispindles prior to the onset of division, but PfARK1 was not expressed in these parasites or in parasites containing inter-polar spindles (Fig. 2b, d). Strikingly, PfARK1 was found specifically associated with mitotic spindles and PfARK1-positive double puncta seemed to be present at the ends of the growing mitotic spindles (Fig. 2b, d). These data strongly indicate that ARK1 is expressed transiently, and only when parasite nuclei are undergoing mitotic division.

These observations were also validated using the rodent malaria parasite, *P. berghei*, a well-established model for studying male gametogony and parasite development within the mosquito host[10]. The subcellular location of PbARK1 in asexual stages was examined by generating an endogenous, C-terminal GFP-tagged PbARK1 (PbARK1-GFP) transgenic parasite line (Supplementary Fig. 2a–c and Supplementary Note 1). Live cell imaging of PbARK1-GFP in asexual blood stages revealed a diffuse location in early ring stages. During trophozoite and early schizont stages, PbARK1-GFP re-localised to discrete focal points in the nucleus (Supplementary Fig. 2d). As schizogony progressed, the focal points of fluorescence divided into two closely associated puncta, which followed subsequent asynchronous nuclear divisions and eventually became diffuse during the later stages (Supplementary Fig. 2d, e). These observations are consistent with the ARK1 location in *P. falciparum* as described above (Fig. 2a).

### PbARK1 is located in the nucleoplasm, at the spindle poles and along microtubules during male gametogony

In gametocytes both before and immediately after activation, PbARK1-GFP had a diffuse distribution predominantly within the nucleoplasm. Within one-min post-activation, part of this was concentrated to a single discrete focal point adjacent to the DNA (Fig. 3a), and then the distribution rapidly extended to form an arc-like structure, which within three-min post-activation divided into two distinct puncta positioned at opposite sides of the nucleus (Fig. 3a and Supplementary Movie 1). The two PbARK1-GFP foci elongated again, forming two arc-like structures along the periphery of the nucleus (Fig. 3a and Supplementary Movie 2). By 6- to 8-min post-activation, a subsequent round of extension and division had produced eight discrete PbARK1-GFP foci, coincident with the completion of three successive rounds of mitosis (Fig. 3a and Supplementary Movie 3). 3D-structured illumination microscopy (SIM) at 1.5–2 min post-activation clearly resolved the arc-like localisation of PbARK1-GFP across the nucleus, with distinct foci visible at both ends of the arc (Fig. 3b).

The arc-like pattern of PbARK1 observed in live cell imaging and 3D-SIM coincided with the pattern of anti-tubulin staining in U-ExM, suggesting an association with the mitotic spindle and spindle poles or inner MTOCs (Fig. 3c and Supplementary Fig. 3a). To further investigate ARK1 location, dynamics, and association with mitotic spindles and kinetochore segregation, we compared the location of PbARK1 with that of the spindle marker ARK2, the kinetochore marker NDC80, and cytoplasmic axonemal protein kinesin-8B. We generated dual fluorescent parasite lines expressing PbARK1-GFP (green) and either PbARK2-mCherry (magenta), PbNDC80-mCherry (magenta), or Pbkinesin-8B-mCherry (magenta) by genetic cross as described previously[4]. Live-cell imaging showed that both PbARK1-GFP and PbARK2-mCherry were co-located at the spindle (Fig. 3d and Supplementary Movie 5). Further 3D-SIM at 1.5–2 min post-activation showed an association of PbARK1-GFP with PbARK2-mCherry at the spindle, with PbARK1-GFP more outside at the spindle poles (Fig. 3e and Supplementary Fig. 3b). In parasite lines expressing PbARK1-GFP and PbNDC80-mCherry, these proteins were located next to the Hoechst-stained DNA, partially overlapping at the spindle but not at the spindle

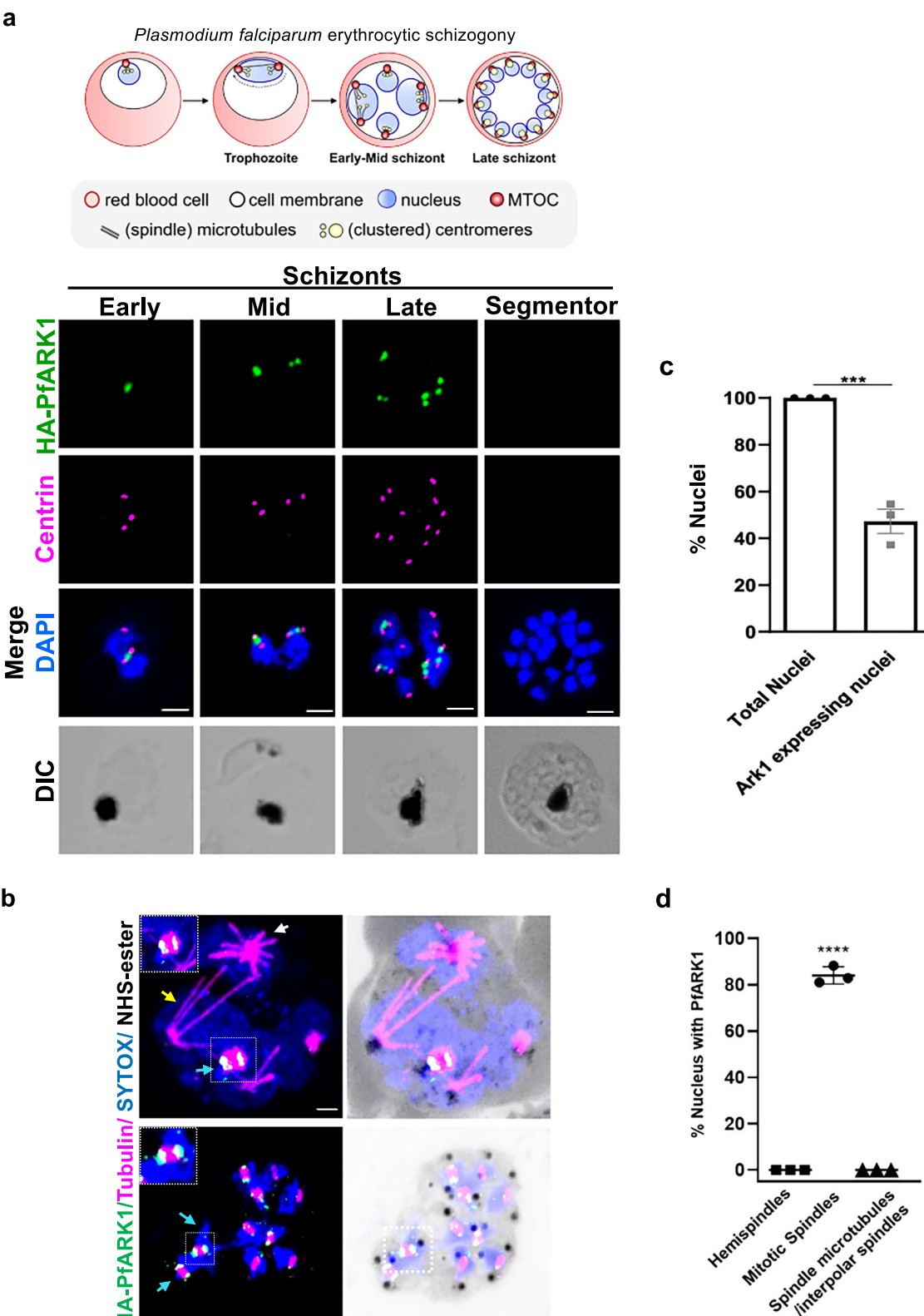

poles (Fig. 3f and Supplementary 4). 3D-SIM at 1.5–2 min post-activation showed a clear association of PbNDC80-mCherry with PbARK1-GFP at the spindle but not at the spindle poles (Fig. 3g and Supplementary Fig. 3c). However, in a parasite line expressing PbARK1-GFP and Pbkinesin-8B-mCherry (a cytoplasmic marker) there was no overlap between these two proteins (Fig. 3h), an observation which was further substantiated by 3D-SIM imaging (Fig. 3i).

During meiotic stages of development post-fertilisation, PbARK1-GFP was diffuse as seen during schizogony, and male gametogony in the cytoplasm, with transient nuclear foci forming in zygotes and disappearing in mature ookinetes (Supplementary Fig. 4a and Supplementary Note 2). A diffuse cytosolic signal with some foci was also observed throughout oocyst and sporozoite stages, including those from salivary glands (Supplementary Fig. 4b and Supplementary

**Fig. 2 | Location of ARK1 during *P. falciparum* asexual division. a** Schematic of asexual division during *P. falciparum* blood stage schizogony. HA-PfARK1-loxP parasites were synchronised and IFA was performed using anti-HA (green) and anti-centrin (magenta) antibodies on schizonts. DNA was stained with DAPI. PfARK1 was observed transiently as a punctum only at nuclei with duplicated centrin foci. Scale bar = 2 μm. **b** Ultrastructure expansion microscopy (U-ExM) was performed on HA-PfARK1-loxP schizonts using anti-HA (green) and anti-tubulin (magenta) antibodies. Hemispindles (white arrow), mitotic spindles (cyan arrow) and interpolar spindles (yellow arrow) that correspond to different stages of nuclear division were observed. PfARK1-positive puncta were mainly associated with mitotic spindles with the puncta flanking the spindle (square and magnification in the inset). Scale bar = 2 μm. **c** Percent nuclei that exhibited PfARK1 staining (panel a) were quantified (SEM ± SE, two-sided unpaired *t* test, ***$P = 0.0005$, $n = 37$ parasites, three independent biological replicates. **d** Nuclei exhibiting PfARK1 foci (panel b) in association with hemispindles, mitotic spindles or interpolar spindles were quantified and represented as % of total nuclei (SEM ± SE, one-way ANOVA, ****$P < 0.0001$, $n > 40$ parasites, three independent biological replicates). Source data are provided as a Source Data file.

Note 2). In liver stages (in HepG2-infected cells), the PbARK1-GFP signal remained diffuse with faint foci even at 48 h post infection (hpi), with slight perinuclear enrichment at 64 hpi (Supplementary Fig. 4c and Supplementary Note 2).

## PfARK1 regulates spindle formation and karyokinesis

The functional role of PfARK1 in asexual blood stage development was assessed by examining the effect of deleting the gene. Rapamycin (RAP) treatment of HA-PfARK1-loxP parasites efficiently excised the floxed sequence (Supplementary Fig. 1c) and caused depletion of PfARK1 protein (Supplementary Fig. 1d). Synchronised ring stage cultures of 1G5DC-SkipFlox (control) (Supplementary Fig. 5a) or HA-PfARK1-loxP parasite lines (Fig. 4a and Supplementary Fig. 5b) were treated with RAP or DMSO (control) and parasite growth was monitored every 24 h. In cycle 0 (RAP-treatment) there was little effect on the parasitaemia of HA-PfARK1-loxP parasites. However, a significant growth defect was observed at the beginning of the next cycle (cycle 1) as indicated by a marked reduction in parasitaemia (Fig. 4a and Supplementary Fig. 5b), whilst there was almost no effect of RAP treatment on 1G5DC-SkipFlox (Supplementary Fig. 5a). This growth defect was further accentuated in subsequent cycles. Microscopic examination of Giemsa-stained blood smears suggested normal ring to trophozoite development in cycle 1 of DMSO-treated parasites (Supplementary Fig. 5c). However, following RAP treatment and PfARK1 depletion, a population of parasites were pyknotic (at ~96 and 144 hpi) and fewer rings were found in cycle 1 and 2 (Fig. 4b and Supplementary Fig. 5c). Schizont development following treatment with E64 to block late schizont maturation and merozoite egress was analysed: a significant number of PfARK1-depleted cycle 1 schizonts had abnormal morphology, with unsegregated nuclei (Fig. 4c and Supplementary Fig. 5d), which is consistent with the role of PfARK1 in nuclear division. This suggestion, that the decrease in parasitaemia resulting from PfARK1 depletion was due to its role in nuclear division and/or schizont maturation, was investigated further.

Next, efforts were made to investigate whether PfARK1 regulates MTOC duplication and spindle formation during asexual division of the parasite. U-ExM revealed the formation of various forms of spindles as described earlier with one or two centrin-labelled foci on the cytoplasmic side of the nuclear envelope (Fig. 4d) and associated with NHS-ester labelled MTOCs or associated with mitotic spindles[23,24]. In PfARK1-depleted parasites, a very few had associated spindles indicated by either absent or speckled tubulin staining (Fig. 4d, grey arrow and Supplementary Fig. 5e) and as mentioned above the nuclei appeared unsegregated. Also, multiple centrin foci were associated with these unsegregated nuclei, and in some cases centrin aggregates were seen away from the nuclei and disorganised (Fig. 4d and Supplementary Fig. 5e, square). Collectively, these data suggested that while the MTOC duplication was unaltered, it was disorganised and spindle formation and karyokinesis were impaired, indicating that PfARK1-depletion leads to defective mitosis.

## PfARK1 depletion impairs segmentation of daughter merozoites

At the end of schizogony, segmentation of the schizont by invagination of the parasite plasma membrane leads to formation of

merozoites prior to their egress[25]. IFAs were performed using antibody to PfMSP1, a parasite surface marker, to assess segmentation following ARK1 depletion. PfMSP1 fluorescence with a bunch of grape-like contours, characterised the membrane of control parasites, but in contrast, PfARK1-depleted parasites either lacked PfMSP1 staining, or contained large aggregations that encompassed multiple merozoites or unsegregated nuclei (Fig. 4e, cyan arrow).

U-ExM of mature (segmented) schizonts with anti-GAP45 antibody-labelling revealed inner membrane complex (IMC) and tubulin labelled subpellicular MT, and rhoptries labelled with NHS-ester (Fig. 4f). However, for the PfARK1-depleted parasites, several unsegregated nuclei were present and these cells either lacked GAP45 staining, or the parasite pellicle (the plasma membrane/IMC) was disorganised (Fig. 4f, cyan arrow, square, zoom and Supplementary Fig. 5f). Although NHS-ester labelled rhoptries were found, their organisation seemed to be disorganised which may be due to aberrant segmentation. In addition, subpellicular MT were either not observed or were mislocalised (Fig. 4f, cyan arrow, square, zoom). These defects were possibly due to arrest in division of the parasites as also indicated by aberrant GAP45 staining. These observations suggested impaired cytokinesis of daughter merozoites upon PfARK1 depletion, the aberrant karyokinesis or nuclear division possibly contributes to these defects.

To study the effect of PfARK1 depletion on merozoite release and egress, time-lapse microscopy was performed[26]. For this purpose, parasites were treated with protein kinase G (PKG)-inhibitor ML-10 to allow schizont maturation but prevent egress, followed by its removal which allowed egress and was captured by time-lapse microscopy[27,28]. Egress and dispersal of merozoites for control parasites was observed (Fig. 4g and Supplementary Movie 6), and while egress from the erythrocyte initially appeared almost normal in PfARK1-depleted parasites, the merozoites either remained attached to the residual body or clumped together (Fig. 4g, magenta arrows, and Supplementary Movie 7, 8) even after an extended period post-egress. These defects likely resulted from the abnormal nuclear segregation and cytokinesis in PfARK1-depleted parasites.

## PbARK1 knockdown reveals an essential role during male gametogony and parasite transmission

Previous analysis by gene disruption had shown that PbARK1 is probably essential for asexual blood-stage development[14]. To be able to manipulate its expression in sexual stages we used a promoter trap strategy, replacing the *ark1* promoter with the promoter from *ama1*, resulting in $P_{ama1}ark1$ hereafter referred as (*ark1PTD*), a gene which is not transcribed in gametocytes but is highly expressed in asexual blood stages (Supplementary Fig. 6a)[29]. Correct genetic integration in the transgenic parasite line was confirmed by PCR (Supplementary Fig. 6b), and *ark1* transcription was shown to be significantly downregulated in *ark1PTD* gametocytes by qRT-PCR (Fig. 5a). A phenotypic analysis of these *ark1PTD* parasites was then performed at different stages of parasite development within the mosquito.

Phenotypic analysis revealed that male gametogony (as scored by exflagellation centres) was severely affected in the *ark1PTD* line compared to GFP control (wild-type-GFP; WT-GFP) parasite (Fig. 5b)[30], and

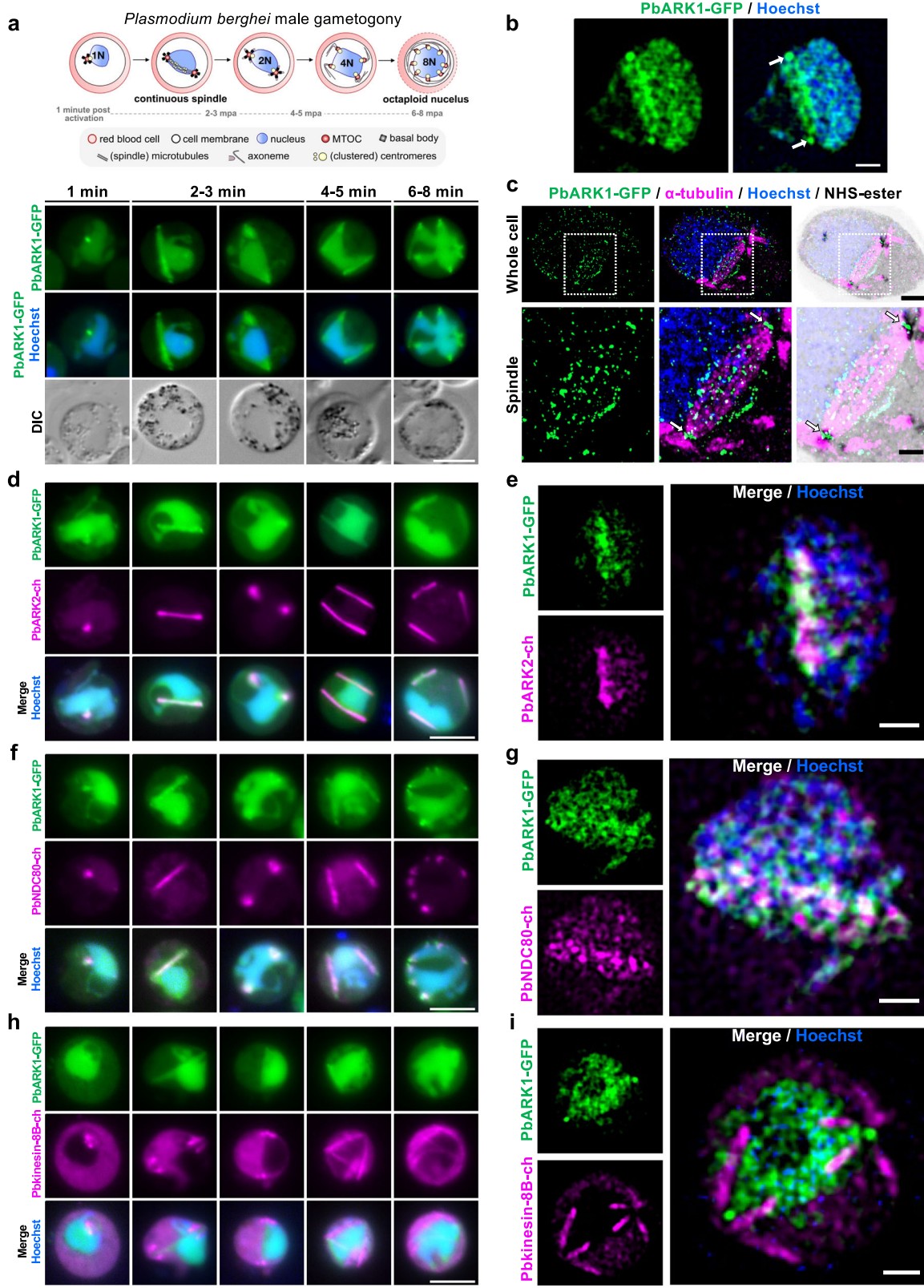

therefore significantly fewer ookinetes (differentiated zygotes after fertilisation) were formed (Fig. 5c). In mosquitoes fed with *ark1PTD* parasites, we observed significantly fewer oocysts in the midgut at day 7 post infection, in contrast to the hundreds of oocysts in mosquitoes fed WT-GFP parasites (Fig. 5d). The number of *ark1PTD* oocysts was further reduced by day 14 and day 21 post infection compared to WT-GFP oocysts (Fig. 5d). The *ark1PTD* oocysts were also significantly

smaller than the WT-GFP oocsts 7 days post infection and diminished further by day 21 post infection (Fig. 5e, f). The number of salivary glands sporozoites was significantly reduced in *ark1PTD* parasites on day 14 (Supplementary Fig. 6c). However, the ratio of salivary gland to midgut sporozoites on day 21 was comparable to that of WT-GFP parasites (Supplementary Fig. 6c), suggesting that the surviving *ark1PTD* sporozoites retain their invasive potential and ability to

**Fig. 3 | Location of ARK1 during *P. berghei* male gametogony.** In all images, PbARK1-GFP is shown in green and DNA stained with Hoechst dye in blue. **a** Top panel: Schematic of male gametogony. Lower panel: Live cell imaging of PbARK1-GFP during male gametogony (1–8 min post activation). DIC: differential interference contrast. Scale bar: 5 μm (n > 50 cells, 5 biological replicates). **b** 3D-SIM of fixed male gametocytes at 1.5–2 min post activation (first spindle stage). Arrows indicate PbARK1-GFP foci. Scale bar = 1 μm (n > 10 cells, 5 biological replicates). **c** U-ExM of male gametocytes at 1.5–2 min post activation (first spindle stage). PbARK1-GFP (green), α-tubulin (magenta), and NHS-ester (grey). Top panels: whole cell. Bottom panels: magnified views of the spindle (white dashed squares). Arrows: PbARK1-GFP foci at the spindle poles (or inner MTOCs). Scale bars = 5 μm (whole cell) and 2 μm (spindle) (n > 8 cells, 2 biological replicates). **d** Live cell imaging of PbARK1-GFP and ARK2-mCherry[10] (magenta) during male gametogony (1–8 min post activation). Scale bar = 5 μm (n > 20 cells, 3 biological replicates). **e** 3D-SIM of fixed male gametocytes at 1.5–2 min post activation (first spindle stage). PbARK1-GFP and ARK2-mCherry. Scale bar = 1 μm (n > 10 cells, 3 biological replicates). **f** Live cell imaging of PbARK1-GFP and NDC80-cherry[61] (magenta) during male gametogony (1–8 min post activation). Scale bar = 5 μm (n > 20 cells, 3 biological replicates). **g** 3D-SIM of fixed male gametocytes at 1.5–2 min post activation (first spindle stage). PbARK1-GFP and NDC80-mCherry. Scale bar = 1 μm (n > 10 cells, 5 biological replicates). **h** Live cell imaging of PbARK1-GFP and kinesin-8B-mCherry[68] (magenta) dynamics during male gametogony (1–8 min post activation). Scale bar = 5 μm (n > 20 cells, 3 biological replicates). **i** 3D-SIM of male gametocytes at 1.5–2 min post activation (first spindle stage) of PbARK1-GFP and kinesin-8B-mCherry. Scale bar = 1 μm (n > 10 cells, 2 biological replicates). Note that the Hoechst signal appears weak in the 3D-SIM image because the focal plane was optimised for the PbARK1-GFP signal located at the apex of the nucleus.

migrate to the salivary glands. To further confirm the absence of sporogony, mosquitoes that had been fed with *ark1PTD* or WT-GFP parasites were allowed to feed on susceptible naïve mice (bite-back experiments). Mosquitoes infected with WT-GFP parasites successfully transmitted the parasite, with blood-stage infection detected in naïve mice after 4 days. In contrast, *ark1PTD* parasites failed to transmit in most experiments (five out of eight) (Supplementary Fig. 6d). However, on three occasions, they were able to establish infection in these bite-back experiments, although the prepatent period was markedly prolonged, with blood-stage parasites first detected 14 days post-infection (Supplementary Fig. 6d). Collectively, these findings indicate that the observed transmission defect is primarily driven by the reduction in sporozoite numbers rather than a loss of sporozoite invasiveness.

To investigate the genome-wide transcriptional impact of PbARK1 depletion, we performed RNA-seq analysis. While few changes were observed in non-activated gametocytes, activated *ark1PTD* gametocytes (30 minute post-activation; mpa) exhibited a broad downregulation of genes involved in signalling, lipid biosynthesis, and ribosome biogenesis. This transcriptional repression is consistent with the observed developmental arrest (Supplementary Fig. 6e-g, and Supplementary Note 3).

### *ark1PTD* parasites have defective spindle formation, MTOC separation, and axoneme formation during male gametogony

To further investigate the function of PbARK1 during male gametogony, WT-GFP and *ark1PTD* male gametocytes were observed during gametogony using U-ExM. In WT-GFP male gametocytes at 8 mpa, individual MTOCs were distinct and separated, whereas in *ark1PTD* male gametocytes multiple MTOCs were clumped and aggregated together (Fig. 5g and Supplementary Fig. 7a). At 15 mpa, WT-GFP male gametocytes were observed undergoing exflagellation, whereas in *ark1PTD* male gametocytes the MTOCs were still clumped together and no exflagellation was observed in the examined cells (Fig. 5g and Supplementary Fig. 7a).

We examined whether the lack of MTOC separation in the *ark1PTD* line was caused by a defect in spindle formation. At 8 mpa, long spindles with clearly separated MTOCs were present in WT-GFP parasites, but in *ark1PTD* lines, short spindles, with the two MTOCs remaining close together (Fig. 6a, b) were present. Measurement showed that *ark1PTD* spindles were significantly shorter (~ 0.8 μm) than WT-GFP spindles (~ 2.5 μm; Fig. 6c). These findings corroborated studies on PfARK1 during schizogony as depletion of the kinase resulted in abnormal spindle formation and accumulation of MTOCs around unsegregated nuclei (Fig. 4d).

TEM observations further revealed structural defects in *ark1PTD* male gametocytes. At 8 mpa, in the *ark1PTD* line, MTOCs formed a normal bipartite structure consisting of an inner spindle MTOC and an outer basal body MTOC, similar to those in WT-GFP male gametocytes, and normal axoneme formation was observed (Fig. 6dA–D, I–L and

Supplementary Fig. 7bA–D), However, we also observed basal body MTOCs that appeared detached from the nucleus in the *ark1 PTD* line (dBB; Fig. 6dI, J and Supplementary Fig. 7bA, B, D). As revealed by expansion microscopy, multiple basal body MTOCs were often aggregated and clumped together (Fig. 6dL). Within the cytoplasm, we observed singlet or doublet MT that may have been axoneme precursors and likely represent failures in axoneme formation (Fig. 6dK, L).

At 15 mpa, WT-GFP gametocytes had started to form gametes with flagellum and nucleus (Fig. 6dE, F, G, H), while in the *ark1PTD* line there was incomplete axoneme formation and exflagellation rarely occurred (Fig. 6dM–P and Supplementary Fig. 7dE–H). In addition, two basal body MTOCs remained close together without proper separation (Fig. 6dM, N, P), with short spindle MT visible between them (Fig. 6dP and Supplementary Fig. 7c). These results suggest that PbARK1 is essential for successful mitosis during male gametogony.

### PbARK1 is part of a divergent chromosome passenger complex (CPC) with two INCENP paralogues

To understand whether ARK1 localisation could be explained by recruitment to specific regulatory scaffolds, we investigated which interacting proteins might direct ARK1 activity and positioning during both asexual and sexual stages of the parasite life cycle. We prepared lysates from *P. berghei* schizonts and male gametocytes expressing PbARK1-GFP, performed GFP-Trap immunoprecipitation, and identified co-purifying proteins by mass spectrometry.

At both stages two divergent inner centromere protein paralogues—here termed INCENP-A and INCENP-B—were the predominant ARK1 interactors (Fig. 7a and Supplementary Fig. 8). This physical interaction is further supported by a recent study demonstrating that PbARK1 co-localises with both INCENP-A and INCENP-B at spindle poles and kinetochores in *P. berghei*[31]. Although replicate numbers were limited (n = 2), ARK1 pulldowns from male gametocytes also showed coordinated enrichment of several spindle and kinetochore proteins amongst the pulldowns (Supplementary Data 2) including components of the ARK2-associated spindle complex (MISFIT-MyoK-EB1, multiple kinetochore proteins (AKiT1, 3, 5 and 7)) and the condensin-II- subunits (CapH2-G2-D3).

No Survivin- or Borealin-like proteins were detected, supporting earlier predictions that *Plasmodium* spp., and more generally apicomplexans, lack these conserved components of the CPC (Fig. 7c). Given the distinct localisation dynamics of ARK1 suggesting that it may not act as the classic passenger complex on chromosomes, it is possible that the chromatin-directing function of Survivin and Borealin has become obsolete for apicomplexans and that the structural motifs for localisation and activation of ARK1 by INCENP-A and -B might be revealed upon detailed comparative sequence analyses.

Because *T. gondii* also encodes two INCENP paralogues that interact with TgARK1[20], we examined INCENP evolution with a focus on alveolates and apicomplexans (Fig. 7b, c). Phylogenetic reconstructions did not reveal a clear scenario supporting either a single ancestral

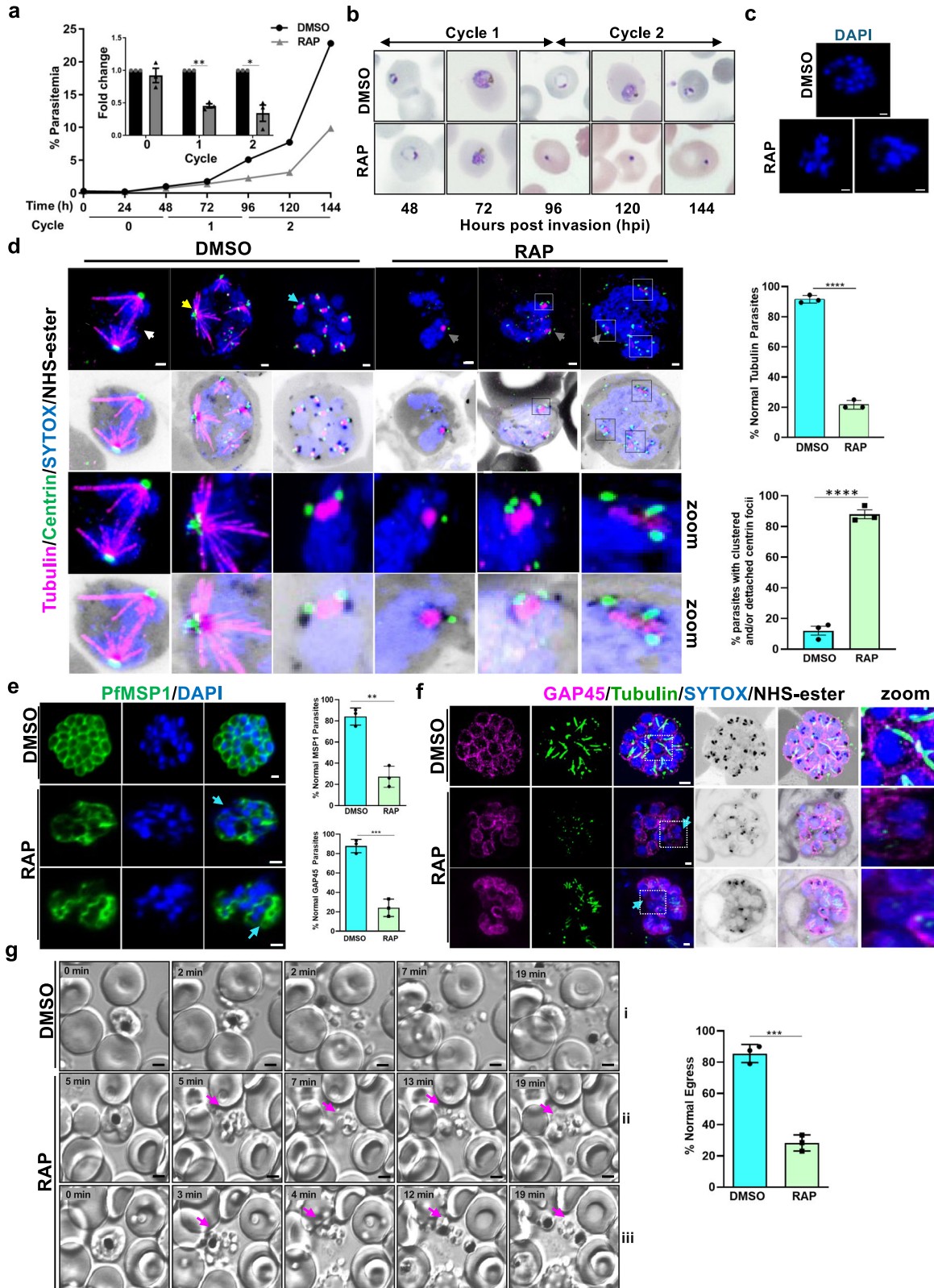

duplication or multiple recurrent duplication events. Instead, the observed patterns are more consistent with parallel, lineage-specific duplications, rather than a conserved INCENP1/INCENP2 duplication inherited from the apicomplexan ancestor—hence we named the paralogue in *Plasmodium* INCENP-A/-B to prevent any misinterpretation of the predicted functional equivalence of INCENP between the *Toxoplasma* and *Plasmodium* paralogues. We found parallel but

separate duplications in other different alveolate lineages such as apicomplexan-like *Squirmida*, *Chrompodellida* and dinoflagellate-related lineages concomitant with the loss of Borealin/Survivin (Fig. 7c), suggesting INCENP duplications might compensate for loss of these chromatin-directing subunits. Notably, INCENP-B is absent from *Piroplasmida*, which also lack ARK2 and ARK3, suggesting functional co-evolution among specific Aurora and INCENP paralogues.

**Fig. 4 | PfARK1 controls karyokinesis, spindle formation and cytokinesis during asexual division. a** Growth assays of synchronised HA-PfARK1-loxP ring stage parasites treated with DMSO or RAP. *Inset*: fold change in parasitemia of RAP-treated parasites relative to control (DMSO) (SEM ± SE, two-sided paired t-test, three independent biological replicates, *P = 0.0345, **P = 0.0035). **b** Giemsa stains of HA-PfARK1-loxP parasites treated with either DMSO or RAP as described in panel (**a**). **c** HA-PfARK1-loxP parasites were treated with DMSO or RAP and cycle 1 schizonts with E64. Parasites ( ~ 40–44 hpi, cycle 1) stained with DAPI (blue). Scale bar = 2 µm. *n* > 70 parasites, three independent biological replicates (see Supplementary Fig. 5d). **d** U-ExM of HA-PfARK1-loxP schizonts ( ~ 40–44 hpi, cycle 1) treated with DMSO or RAP and stained with anti-tubulin (magenta), anti-centrin (green), SYTOX (blue) and NHS-ester (grey). Hemispindles (yellow arrow), mitotic spindles (cyan arrow), interpolar spindles (white arrow). Speckles of tubulin (grey arrows) in RAP-treated parasites are also shown as zoom. Unsegregated nuclei and clusters of centrin (squares) are also shown in zoom. Scale bar = 2 µm. Upper Right Panel: parasites with normal tubulin staining of spindles (SEM ± SE, two-sided unpaired *t* test, ****P < 0.0001, *n* > 55 parasites, three independent biological replicates). Lower Right Panel: % parasites with clustered and/or detached centrin foci (SEM ± SE, two-sided unpaired *t* test, ****P < 0.0001, *n* > 55 parasites, three

independent biological replicates). HA-PfARK1-loxP parasites were treated with DMSO or RAP and cycle 1 schizonts were treated with E64. IFA (**e**) or U-ExM (**f**) was performed on unexpanded (**e**) or expanded (**f**) parasites using anti-MSP1 (**e**, green) or anti-GAP45 (**f**, magenta) and anti-tubulin (**f**, green). RAP-treated parasites either lacked MSP1 and GAP45 staining or contained unsegregated nuclei (**f**, cyan arrow, square, zoom) circumscribed by a large MSP1 contour (**e**, cyan arrow). Middle Panel: Quantification of parasites with normal GAP45 (bottom for panel f) or MSP1 staining (top for panel e) (SEM ± SE, two-sided unpaired *t* test, **P = 0.0016, ***P = 0.0006, *n* > 70 parasites for MSP1 (**e**) and 40 parasites for GAP45 (**f**), three independent biological replicates). Scale bar = 2 µm. U-ExM images are displayed as maximum intensity projection of multiple z slices (**d** and **f**). **g** Time lapse microscopy of HA-PfARK1-loxP cycle 1 schizonts cultured in the presence of DMSO or RAP. Selected still images from videos (Supplementary Movies 6-7) showing normal egress. RAP treated parasites that egressed from erythrocytes but continued to clump together even after prolonged duration are indicated (magenta arrow). Scale bar = 2 µm. Right Panel: parasites undergoing normal egress (SEM ± SE, two-sided unpaired *t* test, ***P = 0.0002, *n* > 100 parasites, three independent biological replicates). Source data are provided as a Source Data file.

AlphaFold3—based co-folding of *Pb*ARK1 with INCENP-A or INCENP-B predicted a conserved IN-box-ARK1 kinase interaction flanked in all apicomplexan proteins by N- and C-terminal extensions that form a beta-sheet—folding INCENP-A/-B around ARK1 (Fig. 7b and Supplementary Fig. 9). In addition, the C-terminal extension has a conserved CxC motif, possibly involved in posttranslational modification (Fig. 7b). We could only predict a coiled-coil in the N-terminus of INCENP-B, and no further structural features could be discerned for INCENP-A using AlphaFold3. Such absence of any feature makes it hard to predict the binding mode of INCENP-A either as a single ARK1-INCENP-A interaction or forming an INCENP-A-ARK1-INCENP-B trimeric complex.

Finally, we examined whether non-canonical sequence features in ARK1 or its INCENP partners might point to alternative models of CPC regulation in *Plasmodium* spp., given that autophosphorylation sites in both Aurora B and INCENP are critical for CPC function in vertebrates[32]. Both the broadly conserved Aurora-kinase activation-loop motif (RxTxCGT) and the canonical INCENP activation site (RTSS in vertebrates) are absent from the *Plasmodium* proteins: ARK1 instead carries a divergent GxSH motif, and INCENP-A and INCENP-B lack an RTSS-equivalent position. However, INCENP-A and INCENP-B retain Aurora consensus motifs in adjacent regions (Fig. 7b, green boxes). For comparison, ARK2 paralogues retain a canonical Aurora activation loop motif, whereas ARK3 possesses a serine-rich activation loop containing multiple candidate phosphorylation sites (Fig. 7c). The absence of clear conserved auto/trans-phosphorylation sites between ARK1 and INCENP-A/-B in *Plasmodium* spp. suggest that CPC kinase activity may rely on a distinct regulatory mechanism, potentially involving input from an additional kinase.

## Discussion

Our cross-stage analysis identified ARK1 as a central organiser of *Plasmodium* mitosis using *P. falciparum* and *P. berghei* model systems. First, ARK1 localises to the spindle and inner MTOC specifically during mitosis, and its depletion in *P. falciparum* reduces spindle formation, prevents proper karyokinesis, disrupts segmentation and impedes egress. Second, during the rapid mitoses of male gametogony, *P. berghei* ARK1 dynamics follow spindle poles; knockdown shortens spindles, block MTOC separation and axoneme biogenesis, and abolishes transmission. These phenotypes demonstrate that ARK1 regulates spindle formation and MTOC separation, thereby ensuring successful sister chromatid segregation and gamete formation.

Our findings show that ARK1 is preferentially expressed and targeted during the mitotic stages marked by spindle formation. Live cell imaging and expansion microscopy revealed that ARK1 locates to the

spindle apparatus and the spindle poles, corresponding to the inner core of the bipartite MTOC[4,23]. Expansion microscopy using MTOC (centrin) and microtubule (tubulin) markers confirmed that ARK1 associates with spindle poles within the inner MTOC in both mitotic stages. Some of these features had been observed earlier for *P. falciparum* ARK1 and further it had been shown that its location is limited to mitotically dividing stages[19]. Dual fluorescence microscopy analysis during male gametogony revealed that it colocalised with ARK2, another aurora kinase reported recently[10], and the kinetochore marker NDC80 at the plus ends of spindle microtubules, suggesting a cooperative role for ARK1 and ARK2 at equatorial spindles. It also has a role in kinetochore dynamics, as observed previously for ARK2[10]. However, live cell imaging in *P. berghei* clearly showed that it is part of the inner MTOC as it does not colocalise with the basal body and axoneme marker, kinesin-8B. This contrasts with *T. gondii* ARK1, which, when tagged at the N-terminus, exhibited a dynamic localisation to both nucleus and cytoplasm during division, but remained predominantly nuclear in non-dividing cells[20,33]. Its colocalisation with kinetochore NDC80 and ARK2 markers potentially supports a dual role in spindle organisation, kinetochore segregation, and spindle pole integrity—reminiscent of Aurora B, which localises to centromeres, the central spindle, and the spindle midzone[5,6,34,35].

Functional genetic studies in *P. falciparum* and *P. berghei* had indicated that ARK1 is likely essential in asexual blood stages[13,14]. Attempts to generate conditional knockdowns using the DD/Shield system in *P. falciparum* were unsuccessful, as reported recently[19]. Here, using the DiCre-Rapamycin system for *P. falciparum*[21] and promoter trap strategy for *P. berghei*[22], we demonstrate that ARK1 depletion leads to defects in spindle formation and karyokinesis. In *P. falciparum*, ARK1 is required for proper spindle elongation, which is critical for karyokinesis. As a result of defects in these processes in PfARK1-depleted parasites there is an impact on subsequent cytokinesis. As a result, parasite growth and dispersal of individual merozoites after egress from RBCs is impaired upon PfARK1 depletion. In *Toxoplasma*, overexpression of a kinase dead mutant of TgARK1 disrupts chromosome segregation and MTOC maturation, similar to some of the division defects observed here in asexual stages, although the mechanism was unexplored[20,33].

During male gametogony, ARK1 knockdown led to defects in the spindle apparatus and the failure of MTOC separation, suggesting its associated function with the outer basal body[4]. This was accompanied by abnormal axoneme formation and very few flagellated male gametes, indicating a failure in MTOC-basal body segregation. These results suggest that while ARK1 is part of the inner MTOC, its functional association with the outer MTOC is essential for successful mitosis and

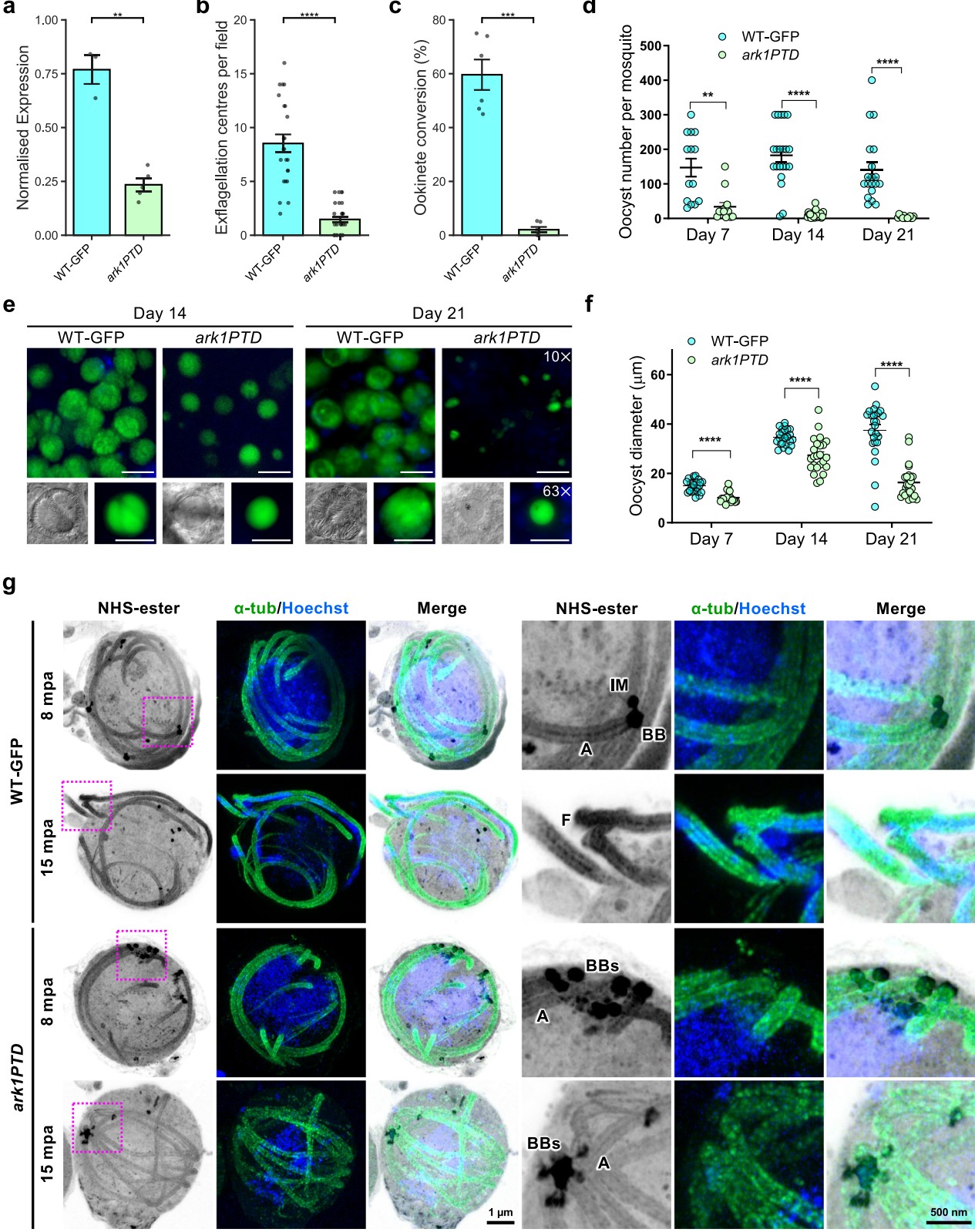

gamete formation. This phenotype is reminiscent of defects reported for other inner MTOC proteins, where the physical connection between the inner and outer plaque components is destabilised[36]. These features also show some resemblance to those of Aurora B in many model systems[5,6,34,35].

Our protein interactome analysis reveals that ARK1 is part of a non-canonical chromosomal passenger complex (CPC), comprising two divergent INCENP paralogues and lacking the canonical chromatin-targeting components Borealin and Survivin. Given that the distinct localisation dynamics of ARK1 suggest that it may not act as the classic passenger complex on chromosomes, it is possible that the chromatin-directing function of Survivin and Borealin has become obsolete for apicomplexans and that the structural motifs for locali-sation and activation of ARK1 by INCENP-A and -B might be revealed

**Fig. 5 | Conditional knockdown of PbARK1 affects male gametogony, disrupting parasite transmission. a** qRT-PCR showing normalised expression of PbARK1 transcripts in WT-GFP and *ark1PTD* parasites. Mean ± SEM. $n = 3$ (WT-GFP) and 5 (*ark1PTD*) independent experiments. **$P < 0.01$ ($P = 0.006$), two-sided Welch's *t* test. **b** Exflagellation centres per field at 15 min post-activation. n = 3 independent experiments (> 7 fields per experiment). Mean ± SEM. ****$P < 0.0001$ ($P = 1.6 \times 10^{-8}$), two-sided Welch's *t* test. **c** Percentage ookinete conversion. $n = 3$ independent experiments (>100 cells). Mean ± SEM. ***$P < 0.001$ ($P = 0.00012$), two-sided Welch's *t* test. **d** Total GFP-positive oocysts per infected mosquito in *ark1PTD* compared to WT-GFP parasites at 7-, 14-, and 21 days post-infection. Mean ± SEM. $n = 3$ independent experiments. Statistical significance was determined using two-sided Welch's *t* tests. **$P < 0.001$, ****$P < 0.0001$. $P = 0.0009$ (Day 7), $1.2 \times 10^{-8}$ (Day 14), and $1.0 \times 10^{-5}$ (Day 21). **e** Midguts at ×10 and ×63 magnification showing oocysts of *ark1PTD* and WT-GFP lines at 14-, and 21 days post-infection ($n > 20$ oocysts, 3

biological replicates). Scale bars: 50 µm (10 ×) and 20 µm (63 ×). **f** The diameter of GFP-positive oocysts in *ark1PTD* compared to WT-GFP parasites at 7-, 14-, and 21 days post-infection. Mean ± SEM. $n = 3$ independent experiments. Statistical significance was determined using two-sided Welch's *t* tests. ****$P < 0.0001$. $P = 2.2 \times 10^{-8}$ (Day 7), $5.0 \times 10^{-5}$ (Day 14), and $1.4 \times 10^{-10}$ (Day 21). **g** Expansion microscopy images of WT-GFP and *ark1PTD* male gametocytes at 8 min or 15 min post-activation (mpa). Three images on the left show maximum intensity projections of whole-cell z-stack images labelled with NHS-ester (grey), α-tubulin (green) and Hoechst (DNA - blue). The three images on the right show magnified views of the areas enclosed by the magenta squares. BB: basal body, BBs: basal bodies (clumped), IM: inner MTOC, A: axoneme. Images are representative of more than 7 cells analysed across 3 independent experiments. The scale bars are scaled by an expansion factor of 4.954. Source data are provided as a Source Data file.

upon detailed comparative sequence analyses. This is consistent with earlier molecular cell biological studies in *Toxoplasma*[20]. While studies in *Toxoplasma* also reported two INCENPs, they did not detect the presence or absence of other CPC components. The lack of chromatin-targeting components might suggest that INCENP-A/-B can facilitate such interactions themselves or simply mean these are lost and replaced by interactions of different proteins. For instance, INCENPs in apicomplexans may form alternative CPC complexes, similar to *Trypanosoma*, where INCENP associates with kinesin-1A/B[37]. A potential candidate kinesin might be kinesin-8X, which shows a similar localisation pattern during different life stages as ARK1[38]. Interestingly, plants and heterokont algae also harbour divergent Survivin-like proteins, with a central conserved but highly divergent helix suggesting a possible broader evolutionary divergence of CPC components across eukaryotes[39]. Future studies will need to include a further scrutiny of INCENP-A/-B localisation and interactomes to determine whether any new CPC components might be present in apicomplexans.

INCENP duplication into INCENP-A and INCENP-B likely arose in parallel to the duplication observed for *Toxoplasma*. Our attempted phylogenetic analyses do not provide evidence to support a clear duplication of INCENPs in the ancestor of both Hematozoa and Coccidia, suggesting that parallel duplications must have occurred in different lineages. Remarkably such recurrent duplications can be observed throughout apicomplexans and consistently coincide with the loss of Survivin (Fig. 7). Notably, INCENP-B has been secondarily lost in piroplasmids, which also lack ARK2 and ARK3, hinting at functional coupling between Aurora paralogues and specific INCENP variants, or pointing towards a loss of meiosis-specific regulators as Roques et al. find that INCENP-B is specific to meiotic stages[31]. Structural predictions using AlphaFold3 suggest that apicomplexan INCENPs contain extended IN-box regions that fold around the ARK1 kinase domain and feature a conserved CxC motif absent from other eukaryotic homologues. At the same time, ARK1 itself shows a divergent activation loop (GxSH instead of the canonical RxT), and INCENPs lack the canonical transphosphorylation site (TSS) site found in other eukaryotes. Such changes might indicate that other kinases are necessary to activate ARK1. Together, these features indicate that apicomplexans have likely rewired CPC architecture and regulation, replacing canonical chromatin-targeting logic with INCENP-centric scaffolds that localise ARK1 at the inner MTOC, spindle and nucleoplasm.

While the function of the other two ARKs (ARK2 and ARK3) is unclear, at least in asexual stages, our recent studies on PbARK2 have provided insights into its role in sexual transmission stages. While ARK2 regulates spindle dynamics during male gametogony just like ARK1 in the present studies, surprisingly, its deletion did not impact exflagellation, unlike that of the ARK1 mutant (Fig. 5b). Previous studies also revealed that PbARK2 deletion caused an increase in the expression of ARK1 and ARK3[10]. Therefore, in the light of the present work, it is reasonable to suggest that ARK1 and ARK2 may have some

overlap of function during male gametogony, particularly in spindle biogenesis. It was also interesting to note that the deletion of PfARK1 resulted in higher PfARK2 levels in schizonts (Supplementary Fig. 1f). While this increase in PfARK2 may be insufficient to fully complement the PfARK1 mutant, it is possible that it may compensate for some of the defects in the growth of PfARK1-depleted parasites. While there may be some overlap in ARK1 and ARK2 function, our results confirm that ARK1 is indispensable for parasite survival, while ARK2 has been previously reported to be essential for asexual development[13,14,40]. A recent study also substantiates that inhibition of PfARK1 results in mitotic defects and microtubule disorganisation during schizogony[16].

Overall, this study suggests that *Plasmodium* ARK1 regulates spindle elongation during mitotic division and associates at the inner nuclear pole of the MTOC. However, its association with the outer MTOC is essential for karyokinesis and for flagellated gamete formation. ARK1 appears to represent a unique Aurora kinase paralogue that, while divergent in structure, shares functional similarities with Aurora B in humans. This suggests that *Plasmodium* has evolved an atypical mechanism of mitosis, where many of the canonical molecules are either missing or highly divergent, reflecting the parasite's adaptation to complex life cycle transitions.

## Methods
### P. falciparum cultures
*Plasmodium falciparum* strain 1G5DC was obtained from European Malaria Reagent Repository (EMRR) and DSM1 was obtained from *BEI* resources, Malaria Research and Reference Reagents Resource Centre (MR4), American Type Culture Collection (ATCC). Parasites were maintained in O+ human erythrocytes (5% haematocrit) in RPMI-1640 supplemented with 0.5% Albumax II and 50 µg/mL hypoxanthine[41]. Cultures were maintained at 37 °C under a mixture of 5% $CO_2$, 3% $O_2$, and 91.8% $N_2$ or 5% $CO_2$. Fresh erythrocytes and culture medium were used to dilute parasites to maintain 3–5% parasitaemia and 5% haematocrit. Relevant drugs were used during the culture of various transgenic lines, as described below. Parasite synchronisation was carried out using 5% sorbitol as described previously[42].

### Generation of transgenic parasite lines
All PCR primers used in this study were synthesized by Sigma and are indicated in Supplementary Data 3.

### 1G5DC SkipFlox
A dimerisable Cre recombinase (diCre) expressing parasite line was generated by transfecting pSkipFlox plasmid, a kind gift from Tobias Spielmann (Addgene Plasmid #85797) into the parent line 1G5DC[21]. Parasites were selected in the presence of 2.5 µg/ml of blasticidin. For simplicity of representation, this parasite line is referred to as 1G5DC in the text and figures.

**HA-PfARK1-loxP** To generate a conditional knockout transgenic parasite, a selection-linked integration (SLI) approach was used. To

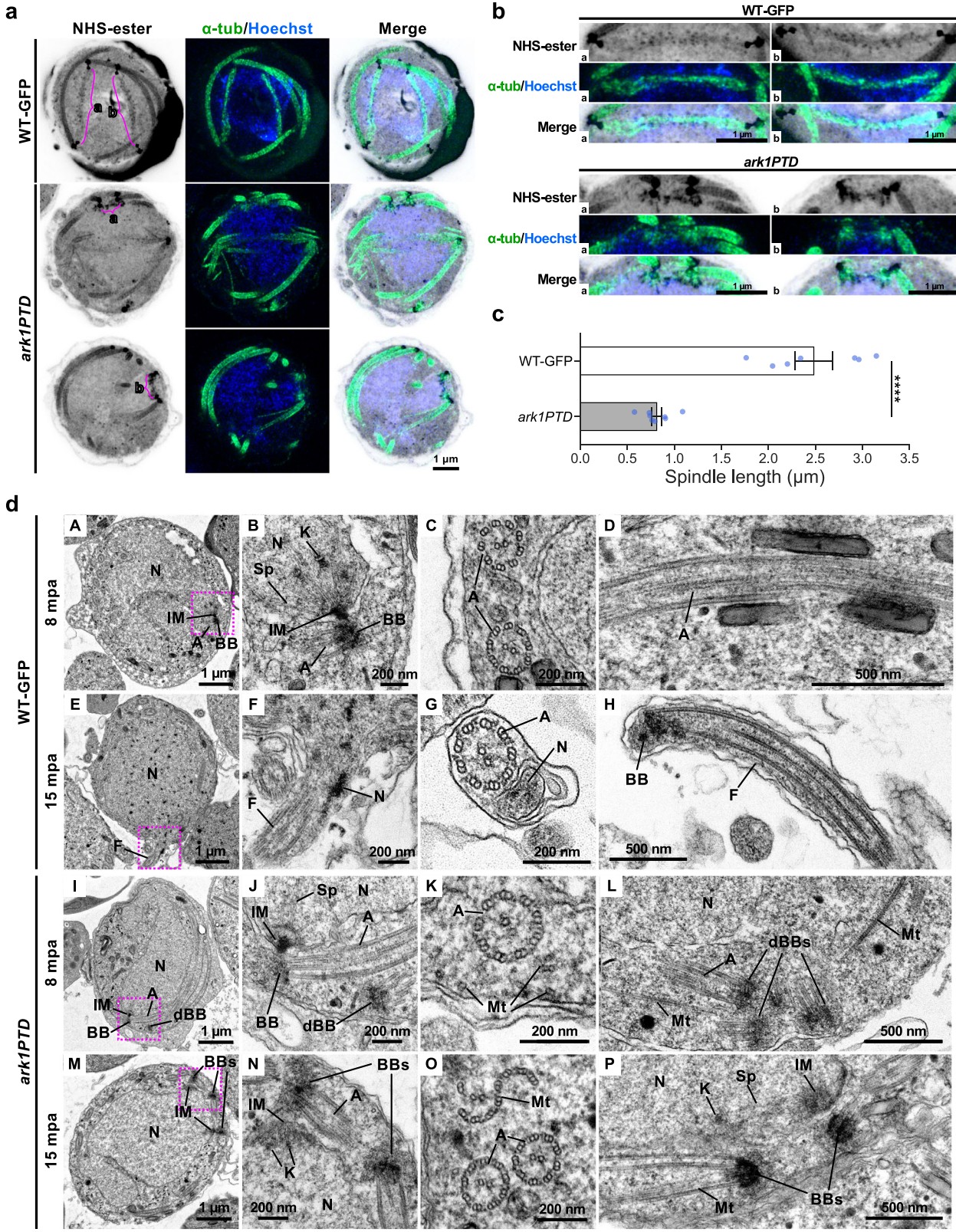

generate the pSLI-HA-ARK1- loxP construct, a 592 bp homology region corresponding to the 5'-end of the PfARK1 gene was amplified using primers (P1 and P2) derived from 3D7 genomic DNA. The PCR product was cloned into *NotI* and *PmeI* sites upstream of loxP and a Myc-tag coding sequence in the pSLI-N-sandwich-loxP (K13) vector, which was a gift from Tobias Spielmann (Addgene Plasmid #85793). A recodonised sequence version of full-length PfARK1 with a N-terminal 3xHA tag and

followed by a loxP site was custom synthesised as a G-block (Genscript). The G-block was cloned in the above construct downstream of yDHODH using *NheI* and *XhoI* restriction sites.

## *P. falciparum* transfection
The pSLI-HA-PfARK1-loxP construct was transfected in 1G5DC SkipFlox parasites at the ring stage using ~100 μg of purified plasmid DNA.

**Fig. 6 | Pbark1 gene-knockdown disrupts spindle and axoneme formation.**
**a** Expansion microscopy images of *P. berghei* wild-type-GFP (WT-GFP) and *ark1* gene-knockdown (*ark1PTD*) male gametocytes at 8 min post-activation (mpa). The maximum intensity projections of entire cells from a z-stack, containing spindles. Cells were labelled with NHS-ester (grey), a-tubulin (green) and Hoechst (blue). Spindles are indicated by magenta lines with (**a** and **b**). Images are representative of more than 7 cells analysed across 3 independent experiments. **b** Magnified views of each spindle of WT-GFP and *ark1PTD* highlighted with magenta lines with a and b in Fig. 6a. **c** Comparison of spindle lengths measured from expansion microscopy data of WT-GFP and *ark1PTD*. Data represent mean ± SEM. (*n* = 7 spindles for WT-GFP and 8 spindles for *ark1PTD*, respectively across 3 different cells). Statistical significance was determined using a two-sided Welch's *t* test. ****$P < 0.0001$

($P = 0.00009$). The scale bars and spindle lengths are scaled by an expansion factor of 4.954, calculated from a comparison of axoneme diameters measured using expansion and transmission electron microscopy images. **d** Transmission electron microscopy images of WT-GFP and *ark1* gene-knockdown male gametocytes at 8 min or 15 min post-activation. B, F, J and N are magnified views of the areas enclosed by the magenta squares in A, E, I and M, respectively. N nucleus, IM inner spindle MTOC, BB basal body, dBB basal body that appeared detached from the nucleus, A axoneme, K kinetochore, Sp spindle microtubule, F flagellum, BBs basal bodies (clumped), Mt microtubule. Images are representative of more than 50 cells analysed across 2 independent experiments. Source data are provided as a Source Data file.

Transgenic parasites were selected by using 2 nM WR99210 and integrants were then enriched using 1.5 μM DSM-1. The parasites that appeared after drug selection were genotyped using PCR primers (Supplementary Data 3 and Supplementary Fig. 1a). The desired integration was confirmed by sequencing of PCR products.

## Conditional knockout of PfARK1
HA-PfARK1-loxP parasites were treated with 250 nM rapamycin at ring stage for 24 h. The genomic DNA was isolated from DMSO/RAP-treated parasites and excision of the PfARK1 locus flanked by loxP sites in the transgenic parasite was confirmed by PCR using primers (P7/P8).

## *P. falciparum* growth rate assays
HA-PfARK1-loxP parasites were cultured in the presence of 10 nM WR99210, 2.5 μg/ml blasticidin and 1.5 μM DSM-1. Cultures were maintained at 2% haematocrit and 7-8% parasitaemia. To perform the growth rate assay, 1G5DC-SkipFlox and HA-PfARK1-LoxP parasites were synchronised at ring stage for two cycles using 5% sorbitol for 20 min at 37 °C, and the rings were seeded at 0.5% parasitaemia and 2% haematocrit. To deplete PfARK1, parasites were treated with either DMSO (vehicle control) or 250 nM rapamycin for 24 h. Thin Giemsa-stained blood smears were examined periodically under a microscope to quantify the parasitaemia at different parasite stages. For assessment of parasitaemia by flow cytometry, cells were fixed by incubation in a solution containing 1% paraformaldehyde and 0.0075% glutaraldehyde at ambient temperature for 15 min on an end-to-end rocker. Post-fixation, samples were either stored at 4 °C or stained immediately with Hoechst 33342 dye at 37 °C for 10 min. After staining, samples were washed with FACS buffer and then analysed on either BDverse (BD Biosciences) or BD FACSymphony A1 Flow Cytometers, acquiring between 10,000 and 100,000 events per sample. Data analysis was carried out using FlowJo software version 10.10.0.

## Purification of *P. berghei* schizonts and gametocytes
Blood-stage parasites were obtained from infected mice on day 4 post-infection and cultured at 37 °C with gentle rotation (100 rpm) for 11 or 24 h. Schizonts were purified the following day using a 60% v/v NycoDenz gradient prepared in phosphate-buffered saline (PBS). The NycoDenz stock solution consisted of 27.6% w/v NycoDenz in 5 mM Tris-HCl (pH 7.2), 3 mM KCl, and 0.3 mM EDTA.

Gametocyte purification was performed by first infecting phenylhydrazine-treated mice[43], followed by sulfadiazine treatment two days post-infection to enrich for gametocytes. Blood was collected on day 4 post-infection, and gametocyte-infected cells were purified using a 48% v/v NycoDenz gradient in PBS, prepared from the same NycoDenz stock solution. Gametocytes were collected from the interface and subsequently activated for downstream assays[44].

## Generation of *P. berghei* transgenic parasites
GFP-tagging constructs were generated using the p277 plasmid vector and transfected into parasites as previously described[45]. A schematic representation of the wild-type *ark1* locus, the GFP-tagging constructs,

and the recombined *ark1* locus is shown in Supplementary Fig. 2a. Correct integration of the *gfp* sequence at the *ark1* locus was confirmed by diagnostic PCR using primer 1 (intArk1tg) and primer 2 (ol492), as indicated in Supplementary Fig. 2b. The expression of native ARK1-GFP was confirmed by Western blot analysis (Supplementary Fig. 2c).

To investigate the function of ARK1, a promoter trap strategy was employed using double homologous recombination to generate a conditional knockdown line (*ark1PTD*). The knockdown construct was derived from the P*ama1* plasmid (pSS368), in which *ark1* was placed under the control of the *ama1* promoter, as previously described[22]. A schematic representation of the endogenous *ark1* locus, the targeting constructs, and the recombined *ark1* locus is shown in Supplementary Fig. 6a. Diagnostic PCR was performed to confirm successful integration, as outlined in Supplementary Fig. 6a. Integration at the 5′ locus was verified using primers intPTD31_5 (Primer 1) and 5′-intPTD (Primer 2), while integration at the 3′ end was confirmed using primers intPTD31_3 (Primer 3) and 3′-intPTama1 (Primer 4) (Supplementary Fig. 6b). Primer sequences are listed in Supplementary Data 3. Transfections were carried out by electroporation using *P. berghei* ANKA line 2.34 for GFP-tagging experiments and ANKA line 507cl1 expressing GFP for generation of the knockdown line[30].

## Generation of dual tagged parasite lines (*P. berghei*)
To generate dual-labelled parasites, ARK1-GFP parasites were mixed in equal proportions with either NDC80-mCherry, ARK2-mCherry, or kinesin-8B-mCherry labelled parasites and co-injected into mice. Four to five days post-infection, when gametocytaemia was high, Anopheles mosquitoes were allowed to feed on the infected mice. Mosquitoes were examined for oocyst development and sporozoite formation at days 14 and 21 post-feeding. Infected mosquitoes were then used to infect naïve mice via mosquito bite-back. Blood-stage infection was assessed 4–5 days later by Giemsa-stained blood smear microscopy. In this way, parasites expressing both ARK1-GFP and either NDC80-mCherry, ARK2-mCherry, or kinesin-8B-mCherry were obtained. Gametocytes from these mixed lines were purified, and fluorescence microscopy was performed as described below to examine protein co-localisation.

## *P. berghei* phenotype analyses
Infections were initiated by intraperitoneal injection of approximately 50,000 *P. berghei* parasites from WT-GFP or *ark1PTD* lines into mice. Asexual stages and gametocyte production were monitored on Giemsa-stained thin smears. Four to five days post-infection, exflagellation and ookinete conversion were assessed using a Zeiss Axiol-mager M2 microscope with an AxioCam ICc1 digital camera. For mosquito transmission assays, 30–50 Anopheles stephensi (SD 500) mosquitoes were allowed to feed for 20 min on anaesthetised, infected mice with ~15% asexual parasitaemia and comparable gametocyte levels (as determined by Giemsa-stained smears). To evaluate midgut infection, ~15 mosquito guts were dissected on day 7, 14 and 21 post-feeding, and oocysts were counted using a 63× oil immersion objective

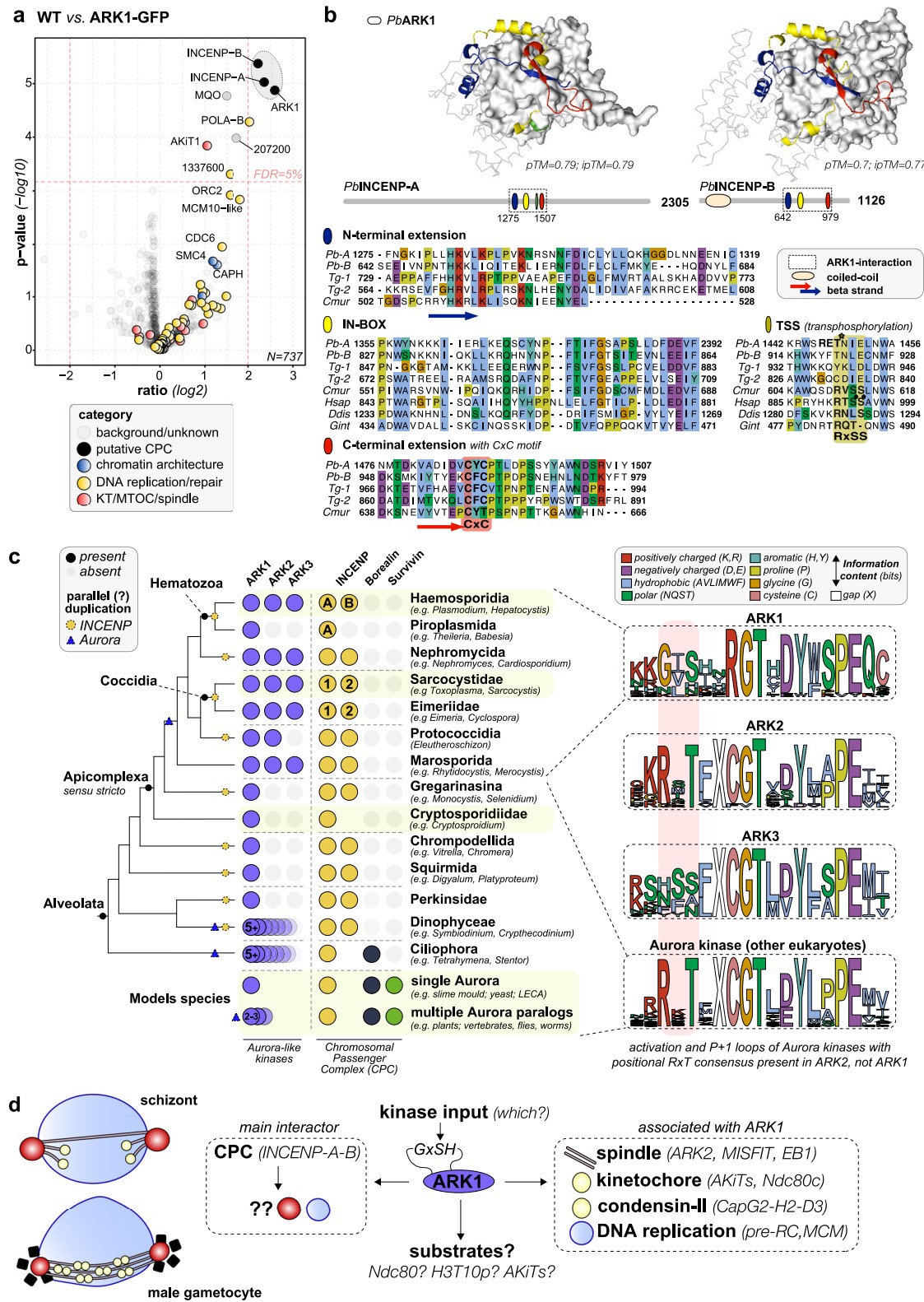

on the same microscope setup. Sporozoites in oocysts were counted on day 14 and day 21 post feeding. Salivary glands were isolated on day 21 to count the sporozoites. Bite-back experiments were performed 21 days post-feeding using naïve mice, and blood smears were examined after 3–4 days. All experiments were conducted at least three times to assess the phenotype.

### Immunofluorescence assay (IFAs)

Immunofluorescence assays (IFA) were conducted on thin blood smears following a previously described protocol[46]. In brief, air-dried smears were fixed for 2 min using ice cold 1:1 mixture of methanol and acetone. The smears were dried to remove any traces of methanol/acetone followed by blocking with 3% BSA for 45 min at room

**Fig. 7 | PbARK1 is the divergent Aurora kinase of the alternative chromosomal passenger complex (CPC) in *Plasmodium* spp. a** Fold enrichment (Log 2 transformed ratio) of normalized iBAQ values of proteins found in co-immunoprecipitates of ARK1-GFP versus control GFP-only for both 8–9 h schizont and 1.5-min male gametocyte lysates (*n* = 2 for all four pulldowns experiments). *P* values were derived using a two-tailed Student's *t* test and corrected for multiple testing using the Benjamini-Hochberg method. False Discovery Rate (FDR) was set at 5%, which corresponds to an unadjusted *P*-value of 0.0067 (see dotted line). The (raw) data underlying this figure can be found in in Supplementary Data 2. **b** Zoom-in on the domain/motif topology of INCENP-A/-B in apicomplexans (Pb = *P. berghei*; Tg = *T. gondii*; Cmur = *C. muris*) and model organisms (Hsap = *Homo sapiens*; Ddis = *Dictyostelium discoideum*; Gint = *Giardia intestinalis*), focused on the extended IN-box region (blue-yellow-red) that interacts with the kinase domain of ARK1. Arrows indicate an antiparallel beta sheet formed by the N- and C-terminal extensions of the IN-box. Green: TSS transphosphorylation site (asterisks indicate (putative) phosphorylated sites) in eukaryotic INCENP homologues, but not in INCENP-A/-B in *P. berghei* (Pb) and *T. gondii* (Tg). RET is a candidate transphosphorylation site in PbINCENP-A (Pb-A). Protein structures are predicted using AlphaFold3, for Predicted Aligned Error (PAE) plot, see Supplementary Fig. 9. **c** Presence/absence of ARK1/2/3, and components of the chromosomal passenger complex amongst the model eukaryotes, the SAR supergroup with a specific focus on Apicomplexa. Lack of phylogenetic signal for one ancestral duplication of INCENP amongst apicomplexans implies that multiple different subgroups underwent independent INCENP duplications. In Plasmodium these paralogues are called INCENP-A and -B; in Toxoplasma these are called INCENP-1 and -2. Right: consensus sequence logos of the activation and P + 1 loop of Aurora kinases in Clustal colour scheme: divergent phosphorylation consensus in ARK1 (GxSH instead of RxT). **d** Summary of findings and discussion points from this study.

temperature. Samples were incubated with primary antibodies for 12 h at 4 °C. After thorough washing with 1X PBS, Alexa Fluor-conjugated secondary antibodies were added and incubated for 2 h at room temperature. Smears were then mounted with VectaShield medium (Vector Laboratories Inc.) containing DAPI for nuclear staining. Microscopy was performed using a LSM980 AR confocal microscope (Carl Zeiss) or Zeiss Axio Observer microscope and image processing was carried out using Zeiss ZEN Blue (versions 3.1 or 3.9). Representative z-stack images were selected for figure preparation unless indicated otherwise. IFA or U-ExM (described below) with MSP1 or GAP45 antibodies was performed to assess segmentation in schizonts that had been treated with 10 μM E64 at ~42 hpi (cycle 1) for 4–5 h to prevent egress and allow complete maturation.

### *P. falciparum* live-cell imaging

Live imaging was carried out with minor modifications to previously described protocols[28]. HA-PfARK1-loxP parasites were synchronised at the ring stage and treated with either DMSO or 250 nM rapamycin for 24 h. Schizonts (cycle 1) were subsequently purified and co-cultured with fresh erythrocytes at 2% haematocrit in RPMI1640 complete medium in the presence of 25 nM PKG inhibitor ML10 [26,27](a gift from LifeArc). ML10 was removed after 4–6 h and parasites were plated onto μ-slide, 2-well ibiTreat chambers (ibidi, 80286) pre-coated with 0.5 mg/mL concanavalin A. The slides were maintained at 37 °C in a humidified chamber with 5% CO$_2$. Imaging was performed using a Zeiss Axio Observer microscope, capturing frames every 3 seconds for a minimum duration of 30 min and the acquired time-lapse videos were processed and analysed using Axio Vision software (version 4.8.2) or ZEN blue 3.1 or 3.9.

### *P. berghei* live cell imaging

To examine PbARK1-GFP expression during erythrocytic development, parasites cultured in schizont medium were imaged at various stages, including ring, trophozoite, schizont, and merozoite. Purified gametocytes were assessed for GFP expression and localisation at multiple time points (0 and 1–15 min) following activation in ookinete medium. For nuclear staining, Hoechst dye was used during live-cell imaging. Zygote and ookinete stages were analysed over a 24-h culture period. Images were acquired using a Zeiss Axio Imager M2 microscope equipped with a 63× oil immersion objective and an AxioCam ICc1 digital camera (Carl Zeiss) using autoexposure settings. Fluorescence signals were adjusted using Axiovision (Rel. 4.8) software to reduce background fluorescence while maintaining detection sensitivity.

### *P. berghei* structured illumination microscopy

Formaldehyde-fixed (4%) gametocytes were stained with Hoechst dye, and 2 μl of the cell suspension was placed on a microscope slide and covered with a 50 × 34 mm coverslip to form a thin, immobilised cell monolayer. Imaging was performed using structured illumination microscopy (SIM) on a Zeiss Elyra PS.1 microscope with either a Zeiss Plan-Apochromat 63×/1.4 oil immersion or Zeiss C-Apochromat 63×/1.2 W Korr M27 water immersion objective, as previously described[36].

### Ultrastructure expansion microscopy

U-ExM was performed largely following previously published protocols[23,47–49]. In brief, to prepare U-ExM samples of *P. falciparum* schizonts, 12 mm coverslips were placed in a 24-well plate and coated with poly-D-lysine (Gibco, A3890401, 0.1 mg/mL) for 1 h at 37 °C, followed by three washes with ultrapure water. *P. falciparum* schizonts were seeded onto the coverslips and incubated for 30 min at 37 °C. Unattached cells were removed gently, and the remaining cells were fixed using 4% (v/v) paraformaldehyde in PBS for 15 min at 37 °C. After fixation, coverslips were rinsed with prewarmed PBS and incubated for 12 h at 37 °C in 1.4% (v/v) formaldehyde and 2% (v/v) acrylamide in PBS. A monomer solution consisting of 19% (w/w) sodium acrylate (Sigma-Aldrich, 408220), 10% (v/v) acrylamide (Sigma-Aldrich, A3553), and 2% (v/v) N,N'-methylenebisacrylamide (Sigma-Aldrich, M7279) in PBS was prepared 24 h in advance and stored at −20 °C. For polymerisation, 5 μL of 10% (v/v) TEMED (Sigma-Aldrich, T9281) and 5 μL of 10% (w/v) APS (Sigma-Aldrich, A3678) were mixed with 90 μL of the thawed monomer solution. The gel mix (35 μL) was pipetted onto parafilm squares pre-cooled at −20 °C and the coverslip was placed over the drop. Gels were allowed to polymerise for 1 h at 37 °C. Polymerised gels were transferred to denaturation buffer (200 mM SDS, 200 mM NaCl, 50 mM Tris, pH 9) and shaken for 15 min at room temperature, then incubated at 95 °C for 90 min. Denatured gels were expanded by sequential incubations in water (30 min each, repeated thrice) with gentle shaking. Expanded gels were equilibrated in PBS for 15 min and blocked in 3% BSA/PBS for 30 min at room temperature. Incubation with primary antibody (details in Supplementary Data 4) was carried out for 12 h at room temperature on a shaker, followed by three 10 min washes with PBST. Gels were then incubated with secondary antibodies, 405 NHS Ester (Thermo Fisher, A30000) and SYTOX/Hoechst in PBS for 2.5 h in the dark with gentle shaking followed by three PBST washes and final equilibration in MilliQ water (30 min, repeated thrice).

For imaging, μ-slide 2-well ibiTreat chambers (ibidi, 80286) were coated with poly-D-lysine (0.5 mL, 1 h at 37 °C), rinsed with MilliQ water, and used to mount the gels cell side down. Imaging was performed using a Zeiss LSM980 confocal microscope with a 63X oil immersion objective (NA 1.4). Z-stacks (0.5 μm intervals) were captured and processed using ZEN Blue (versions 3.1 or 3.9). Deconvolution was performed using Zen Deconvolution tool kit. U-ExM images are displayed as maximum intensity projections of full z-slices unless indicated otherwise.

To prepare U-ExM samples of *P. berghei* male gametocytes, purified and 8 min- and 15 min-activated gametocytes were fixed in 4% formaldehyde in PHEM buffer (60 mM PIPES, 25 mM HEPES, 10 mM EGTA, 2 mM MgCl$_2$, pH6.9) at room temperature for 15 min. Fixed samples were attached to 10 mm round Poly-D-Lysine coated coverslips for 15 min. Coverslips were incubated overnight at 4 °C in 1.4%

formaldehyde (FA)/ 2% acrylamide (AA). Gelation was performed in ammonium persulfate (APS)/TEMED (10% each)/monomer solution (23% sodium acrylate; 10% AA; 0,1% BIS-AA in PBS) on ice for 5 min and at 37 °C for 30 min. Gels were denatured for 15 min at 37 °C and for 90 min at 95 °C in denaturation buffer (200 mM SDS, 200 mM NaCl, 50 mM Tris, pH 9.0, in water). After denaturation, gels were incubated in distilled water overnight for complete expansion. The following day, circular gel pieces with a diameter of ~13 mm were excised, and the gels were washed in PBS three times for 15 min to remove excess water. The gels were then incubated in blocking buffer (3% BSA in PBS) at room temperature for 30 min, incubated with mouse monoclonal anti-a-tubulin antibody (T9026, Sigma-Aldrich, 1:500 dilution) and/or anti-GFP antibody (A11122, Invitrogen, 1:500 dilution) in blocking buffer (1:500 dilution) at 4 °C overnight and washed three times for 15 min in wash buffer (0.5% v/v TWEEN-20 in PBS). The gels were incubated with 8 µg/ml Atto 594 NHS ester (08741, Merck), 10 µg/ml Hoechst 33342 (H1399, Invitrogen), Alexa Fluor 488 goat anti-mouse IgG (A11001, Invitrogen, 1:500 dilution) and Alexa Fluor 568 goat anti-mouse IgG (A11004, Invitrogen, 1:500 dilution) in PBS at 37 °C for 3 h followed by three washes of 15 min each in wash buffer (blocking and all antibody incubation steps were performed with gentle shaking). The gels were then washed three times for 15 min with wash buffer and expanded overnight in ultrapure water. The expanded gel was placed in a 35 mm glass bottom dish (MATTEK) with the 14 mm glass coated with Poly-D-Lysine and mounted with an $18 \times 18$ mm coverslip to prevent the gel from sliding and to avoid drifting while imaging. High-resolution confocal microscopy images were acquired using a Zeiss Celldiscoverer 7 with Airyscan using a Plan-Apochromat 50 ×/1.2NA Water objective, with 405, 488 and 561 nm lasers. Confocal z-stacks were acquired using line scanning and the following settings: $55 \times 55$ nm pixel size, 170 nm z-step, 2.91 µs/pixel dwell time, 850 gain and 3.5% (405 nm), 4.5% (488 nm) and 5.0% (561 nm) laser powers. The z-stack images were processed and analysed using Fiji (Version 1.54 f)[50].

### Calculation of the expansion factor for the U-ExM of male gametocytes
The expansion factor for the U-ExM of male gametocytes was calculated by comparing axoneme diameters from two types of images (U-ExM and TEM). The diameters were measured using Fiji (Version 1.54 f)[50]. The average diameter of the axoneme in NHS-stained U-ExM images of male gametocytes activated for 8 min was 933.83 nm ($n = 12$), while the average diameter in TEM images of male gametocytes activated for 8 min was 188.50 nm ($n = 8$). We used the calculated expansion factor of 4.954 for the analysis.

### Electron microscopy of male gametocytes
*P. berghei* gametocytes activated for 8 min and 15 min were fixed in 3% glutaraldehyde in 0.1 M cacodylate buffer and processed for electron microscopy[51]. Briefly, samples were post-fixed in 1% aqueous osmium tetroxide, treated en bloc with 2% aqueous uranyl acetate, dehydrated in graded ethanol series and embedded in Spurr's epoxy resin. Thin sections were stained with lead citrate prior to examination in a Tecnai G2 12 BioTwin (FEI UK, UK) or a Jeol JEM-1400Flash (JEOL, Japan) electron microscope.

### Liver stage parasite imaging
For *P. berghei* liver-stage imaging, 100,000 Hepatocelluar carcinoma cells (HepG2) were seeded onto glass coverslips in 48 well plate. Salivary glands from *Anopheles stephensi* mosquitoes infected with PbARK1-GFP parasites were dissected and homogenised using a pestle to release sporozoites. The sporozoites were gently pipetted onto the HepG2 cell monolayer and incubated at 37 °C with 5% $CO_2$ in complete minimum Eagle's medium The culture medium was replaced 2 h post-infection and subsequently changed daily. For live-cell imaging,

Hoechst 33342 (Molecular Probes) was added to a final concentration of 1 µg/ml to stain host and parasite nuclei. Infected cells were imaged at, 48-, and 64 -hours post-infection using a Zeiss Axio Imager M2 microscope equipped with a 63× oil immersion objective and an AxioCam ICc1 digital camera (Carl Zeiss) using autoexposure settings. and Leica Application Suite X software. Each experiment was performed in triplicate, and 20–30 infected cells were analysed per time point to assess PbARK1-GFP localisation.

### Immunoblotting
*P. falciparum*—Immunoblotting was performed as described previously[52] using primary antibodies (Supplementary Data 4) and HRP-conjugated secondary antibodies and protein bands were visualized using either SuperSignal™ West Pico or Femto chemiluminescent substrates (Pierce, USA) using X-ray films.

*P. berghei* - Purified gametocytes were lysed in buffer containing 10 mM Tris-HCl (pH 7.5), 150 mM NaCl, 0.5 mM EDTA, and 1% NP-40. After lysis, Laemmli buffer was added, and samples were boiled for 10 min at 95 °C, followed by centrifugation at $13,000 \times g$ for 5 min. The supernatants were separated on a 4–12% SDS-polyacrylamide gel and transferred onto a nitrocellulose membrane (Amersham Biosciences). Immunoblotting was performed using the Western Breeze Chemiluminescence Anti-Rabbit Kit (Invitrogen), following the manufacturer's protocol. An anti-GFP polyclonal antibody (Invitrogen) was used at a 1:1,250 dilution to detect PbARK1-GFP.

### Quantitative real-time PCR (qRT-PCR) analyses
*P. falciparum*—Total RNA was extracted using TRIzol reagent (G Biosciences) followed by purification with the Qiagen RNeasy Mini Kit (Cat. No. 74104). cDNA synthesis was performed using RevertAid H Minus Reverse Transcriptase (Thermo scientific, EP0451) and 1 µg of DNase-treated RNA as the template. Quantitative PCR was conducted using the CFX96 Real-Time PCR Detection System (Bio-Rad). Reactions were normalized using 18S rRNA as the internal control. Relative gene expression levels were calculated using the $2^{-\Delta\Delta Ct}$ method. Primer sequences used for the qRT-PCR are listed in Supplementary Data 3.

*P. berghei* - Total RNA was extracted from gametocytes using the RNA Purification Kit (Stratagene), and cDNA was synthesised using the RNA-to-cDNA Kit (Applied Biosystems). qRT-PCR was performed with cDNA from 80 ng of RNA using SYBR Green Fast Master Mix (Applied Biosystems) on an Applied Biosystems 7500 Fast system. Cycling conditions were: 95 °C for 20 s, followed by 40 cycles of 95 °C for 3 s and 60 °C for 30 s. Primers were designed using Primer3 (https://primer3.ut.ee/). Each gene was tested in three biological and technical replicates. *hsp70* (PBANKA_081890) and *arginyl-tRNA synthetase* (PBANKA_143420) were used as reference genes. Primer sequences are listed in Supplementary Data 3.

### Transcriptome analysis using RNA-seq
Total RNA was extracted from activated gametocytes and schizonts of *P. berghei* WT-GFP and *ark1PTD* parasites (three biological replicates each) using the RNeasy Kit (Qiagen), vacuum-concentrated, and transported in RNA-stable tubes (Biomatrica). Strand-specific mRNA libraries were prepared with the TruSeq Stranded mRNA Sample Prep Kit (Illumina) and sequenced on an Illumina HiSeq 4000 platform (paired-end, 150 bp reads). Read quality was assessed with FASTQC, and trimming of low-quality reads and adapter sequences was done using Trimmomatic[53]. Reads were mapped to the *P. berghei* ANKA genome (PlasmoDB release 40) using HISAT2 v2.1.0[54], and gene counts were generated with FeatureCounts[55]. Low-expression genes (CPM < 1) were excluded. Data were normalized using the TMM method (EdgeR[56]), transformed with voom (limma[57]), and analysed for differential expression using DESeq2[58]. Statistical significance was determined using a two-sided Wald test, with P-values adjusted for multiple comparisons

using the Benjamini-Hochberg procedure. Genes with fold-change > 2 and FDR < 0.05 were considered differentially expressed.

## Immunoprecipitation and mass spectrometry

*P. berghei* purified schizonts and gametocytes (activated for 1.5–2 min) were crosslinked using formaldehyde 10-min incubation with 1% formaldehyde, followed by 5-min incubation in 0.125 M glycine solution and 3 washes with PBS (pH 7.5). Immunoprecipitation was performed using crosslinked protein and a GFP-Trap_A Kit (Chromotek) following the manufacturer's instructions. Proteins bound to the GFP-Trap_A beads were digested using trypsin and the peptides were analysed by LC-MS/MS and subsequently visualised using both Principal Component Analysis (PCA) and for relative ratios of WT/POI using a volcano plot as published previously[36,59]. In short, significance of enrichment was assessed using the two-tailed Student *t* test (implemented in R), and *p* values were corrected for multiple testing using the Benjamini–Hochberg method. iBAQ values were normalized by the sum of all iBAQ values over one experiment. Only those proteins were considered with two or more unique peptides and at least 2 values in ARK1 pulldowns. Volcano plots were visualised using VolcaNoseR[60].

## Evolutionary bioinformatics

Our predicted proteome set was based on a version previously used (see Supplementary Data 5)[61]. Phyletic profiles for ARK1-3 and INCENP were partially derived from previous studies[10,62]. Sequence searches for missing orthologues were performed with the HHPred webserver from MPI-toolkit[63,64] and the hmmer package[63], as previously described[61]. For the larger gene families like Aurora kinases, we used IQ-tree-based maximum-likelihood phylogenetics (standard setting - model finding, 1000 bootstraps[65]) to check the validity of bidirectional-best-blast (BBH) hits amongst our genome sets on an ad hoc basis. Off note, we could not reliably generate a single phylogenetic tree harbouring all three ARKs within the Aurora kinase orthologous group (OG), suggesting that ARK2 and ARK3 are either extremely divergent or not Aurora kinases. Similarly, for the INCENP gene family, multiple alignments were generally too divergent and phylogenetically informative stretches too short to infer any consistent phylograms. All sequences used for this study can be found in Supplementary Data 5.

## AlphaFold3 modelling

Co-folds of 3D protein structures for PbARK1 and the extended IN-box of either PbINCENP-A or -B were modelled using the AlphaFold3 webserver (https://alphafoldserver.com/) with standard settings (seed set to 100)[66].

## Ethics statement

All animal procedures were approved following an ethical review process and conducted in accordance with the United Kingdom Animals (Scientific Procedures) Act 1986, under Home Office Project Licenses (PDD2D5182 and PP3589958). Female CD1 outbred mice (6–8 weeks old) were obtained from Charles River Laboratories and used for all experiments carried out in the UK. The conditions of mice kept are a 12 h light and 12 h dark (7 till 7) light cycle, the room temperature is kept between 20 and 24 degrees Celsius and the humidity is kept between 40 and 60%.

## Reporting summary

Further information on research design is available in the Nature Portfolio Reporting Summary linked to this article.

## Data availability

The RNA-seq data generated in this study have been deposited in the NCBI Sequence Read Archive under accession number PRJNA1309997. The mass spectrometry proteomics data generated in this study have been deposited in the ProteomeXchange Consortium via the PRIDE[67] partner repository under the dataset identifier PXD068821 and 10.6019/PXD068821 [https://www.ebi.ac.uk/pride/archive/projects/PXD068821]. All other Source data are provided with this paper.

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

## Acknowledgements

R.T. is supported by ERC advance grant funded by UKRI Frontier Science (EP/X024776/1), MRC UK (MR/K011782/1), and BBSRC (BB/L013827/1, BB/X014681/1). M.Z., D.B., and A.M. were supported as research fellows by (EP/X024776/1). R.Y. and D.S.G. are supported by BBSRC (BB/X014681/1). S.L.P. is supported by Wellcome DBT India Alliance/Team Science (IA/TSG/21/1/600261). A.A.H. is supported by the Francis Crick Institute (FC001097), which receives core funding from the Cancer Research UK (FC001097), the UK Medical Research Council (FC001097), and the Wellcome Trust (FC001097). KGLR is supported by the NIH/NIAID (R01 AI136511) and the University of California, Riverside NIFA-Hatch-225935. ECT was supported by a personal fellowship from the Nederlandse Organisatie voor Wetenschappelijk Onderzoek (NWO), the Netherlands (grant no. VI. Veni.202.223). Studies on *P. falciparum* were also supported by a Team Science grant (IA/TSG/21/1/600261) to PS, RT, and TSKP from DBT-Wellcome Trust India Alliance and funds from NII core. PS is a recipient of J.C. Bose Fellowship. A.E. was supported by a Commonwealth Academic Fellowship awarded by the Commonwealth Scholarship Commission in the UK. We thank the School of Life Sciences Imaging (SLIM) for providing access to confocal and SIM microscopy and Dr Robert Markus for technical assistance. Most microscopy on *P. falciparum* was done in National Institute of Immunology (NII)-confocal microscopy facility. We thank the Nanoscale and Microscale Research Centre (nmRC) for providing access to instrumentation and Dr Michael W. Fey, Dr Julie Watts and Ms Nicola J. Weston for technical assistance. We are grateful to Dr Sabrina Absalon (Indiana University) for technical advice on U-ExM. For Open Access, the authors have applied a CC BY public copyright licence to any Author Accepted Manuscript version arising from this submission. We thank Cleidiane Zampronio at Warwick University for mass spectrometry methods and Bio Support Unit, University of Nottingham for maintenance of mice used in this study.

## Author contributions

R.T., P.S., and E.C.T. conceptualised the project. A.N. and P.S. performed the *P. falciparum* experiments. R.Y., M.Z., S.L.P., A.M., E.R., and R.T. performed the *P. berghei* live-cell imaging. R.Y., D.J.F., and S.V. performed the TEM observation. R.Y. and M.Z. performed the *P. berghei* U-ExM observation. M.Z., A.E., and D.B. performed the qPCR experiment. S.A. and K.G.L.R. performed the RNAseq analysis. R.T., D.B., A.R.B., and E.C.T. performed the proteomics analysis. E.C.T. performed the evolutionary bioinformatics analysis. A.N., R.Y., M.Z., D.J.F., S.L.P., A.M., A.R.B., S.V., K.G.L.R., D.S.G., A.A.H., E.C.T., P.S., and R.T. performed formal analysis. A.N., R.Y., M.Z., D.J.F., S.A., S.L.P., A.M., E.C.T., and R.T. performed visualisation. R.T., P.S., and E.C.T. acquired funding. R.T., P.S., and E.C.T. supervised the project. A.N., R.Y., M.Z., A.A.H., E.C.T., P.S., and R.T. wrote the manuscript draft. All the authors reviewed and edited the manuscript.

## Competing interests

The authors declare no competing interests.

## Additional information

¹Eukaryotic Gene Expression Laboratory, National Institute of Immunology, New Delhi, India. ²School of Life Sciences, University of Nottingham, Nottingham, UK. ³Department of Biological and Medical Sciences, Oxford Brookes University, Oxford, UK. ⁴Department of Molecular, Cell and Systems Biology, University of California Riverside, Riverside, CA, USA. ⁵School of Life Sciences, Gibbet Hill Campus, University of Warwick, Coventry, UK. ⁶Malaria Parasitology Laboratory, The Francis Crick Institute, London, UK. ⁷Cell Biochemistry, Groningen Biomolecular Sciences and Biotechnology Institute, Faculty of Science and Engineering, University of Groningen, Groningen, The Netherlands. ⁸Present address: Division of Molecular Microbiology and Immunology, CSIR-Central Drug Research Institute, Lucknow, India. ⁹These authors contributed equally: Annu Nagar, Ryuji Yanase, Mohammad Zeeshan. ¹⁰These authors jointly supervised this work: Eelco C. Tromer, Pushkar Sharma, Rita Tewari. ✉e-mail: e.c.tromer@rug.nl; pushkar@nii.ac.in; rita.tewari@nottingham.ac.uk

