## [Transparent Peer Review file · Nature Communications]

***Plasmodium* ARK1 regulates spindle formation during atypical mitosis and forms a divergent chromosomal passenger complex**

Corresponding Author: Professor Rita Tewari

Version 0:

Reviewer comments:

Reviewer #1

(Remarks to the Author)

The manuscript by Nagar and colleagues evaluates the function of ARK1 in *Plasmodium* parasites. Using the rodent malaria, *Plasmodium berghei*, and the human *Plasmodium falciparum*, ARK1 is hypothesized to be part of the inner MTOC and to influence spindle/kinetochore dynamics. The findings are most striking during male gamete formation.

The manuscript combines elegant genetic, cell biology, and evolutionary biology to understand the function of the divergent ARK1 in *Plasmodium*. This is a nice study that combines findings in two *Plasmodium* species. The findings are largely well-supported. I have suggested some additions and changes that I think will improve the study.

Major

1. Claims about the timing of ARK1 expression in line 124-125 are likely true but are only weakly supported by the provided data of two schizonts. It would be helpful for the reader to have some quantification of this phenotype. One could imagine counting the number of nuclei in each stage of their karyokinesis cycle and noting if ARK1 is present or absent. I do not doubt the result shown, but it would be stronger with quantification. Line 127-137, please explain why the localization of PbARK1 is diffuse when this localization is not mentioned or observed for PfARK1. This is likely due to method differences, but the claim that the localization is the similar is not fully supported without further explanation.
2. The RNAseq paragraph (lines 279-300) does not seem to add any useful information. I think this is distracting and detracts from the more interesting story that is unfolding. This should be removed because the observations/hypotheses are not experimentally tested.
3. The use of immunoprecipitation to identify INCENP1 and INCEP2 is a strong piece of data. It would be nice, if possible, to show colocalization of ARK1 and these proteins. While it seems extremely likely that they will colocalize, it is good technique to validate findings from mass spectrometry with additional experiments. Because these structures are stable and associated with microtubules, they may “interact” or immunoprecipitated over large distances.

Minor

1. Figure 1b is a nice figure to give the reader a map of the proposed localization and function. The small blue “nucleoplasm” dot is a bit odd and was confusing for me. I think that this would be better to move out of the center of the panel and into a legend box or something. This extra dot seems like it should be a structure when it is not really anything.
2. Line 123 – the term “polar ends” should be clearly defined.
3. Fig 2E – the GFP antibody staining has high background. Can a different cell be selected? Perhaps a different gel preparation and a different antibody preparation (or source of antibody) will help to reduce this unfortunate background.
4. Line 197 – the statement “a significant number” needs to be backed up by a statistical measurement or removed/changed.
5. Line 234-236 – The primary defect is abnormal karyokinesis. The cytokinesis defect is likely entirely secondary to this primary defect. It should be stated very explicitly that ARK1 has little to do with cytokinesis. While this is implied, for the non-expert reader it should be stated clearly.
6. In my opinion, the fig 6b could be moved to a supplement. The AlphaFold3 server is publicly available and anyone could generate these models on their own.

Reviewer #2

(Remarks to the Author)

Summary:

In this manuscript, Nagar and colleagues present a comprehensive investigation of Aurora-related kinase 1 in *P. falciparum* schizogony and *P. berghei* male gametogony, combining live-cell imaging, expansion microscopy, and electron microscopy. The authors demonstrate that PbARK1 is essential for spindle formation, MTOC separation, exflagellation, and parasite transmission, while PfARK1 disruption impairs spindle assembly and normal karyokinesis/cytokinesis. Proteomics analysis suggests that PbARK1 interacts with INCENP1/2 to form a chromosome passenger complex.

Overall, the study provides valuable insight into ARK1 function in both asexual and sexual stages of Plasmodium, contributing to a better understanding of how ARK1 regulates cell division through distinct mitotic mechanisms in schizogony and gametogony. However, several methodological and data presentation issues need to be addressed. Specially, the internal HA-tag insertion in PfARK1 requires further justification. Additionally, the PfARK1 expansion microscopy images lack sufficient resolution or stage comparability to convincingly support claims that PfARK1 deficiency causes centriolar plaque mis-segregation or separation from the nucleus, as well as altered subpellicular microtubule polymerization or localization. Therefore, additional clarification and higher-resolution data are required before publication.

Major comments:

1) Line 112 and Extended Data Fig. 1a: The authors describing generating a HA-PfARK1-loxP parasite line, which appears to be N-terminally HA-tagged. However, Extended Data Fig. 1a shows that the HA tag was inserted internally within the PfARK1 coding sequence rather than at the N- or C-terminus. Although the localization pattern resembles PfARK1-GFP reported previously, and this line was used for conditional PfARK1 disruption, it remains unclear whether this internal tagging affects the protein's native function or folding. Please clarify the rationale for this strategy and specify whether the HA tag lies in an unstructured or non-conserved region.

2) Lines 113-125 and Fig. 2a-b: PfARK1 localization has been reported previously (PMID: 21166904; PMID: 39235260). The data here are confirmatory and should be stated as consistent with those earlier findings. For Fig. 2a-b, please provide higher-magnification PfARK1/Tubulin/NHS-ester merged images to improve clarity and data presentation.

3) Fig. 3d: Please provide higher-magnification merged images combining the tubulin, centrin, SYTOX, and NHS-ester channels to better illustrate the described phenotype. Current overview images lack resolution to convincingly support the statement that "the nuclei appeared unsegregated with associated multiple centrin foci, and centrin aggregates were seen away from the nuclei and disorganized" (line 211-212). In addition, the tubulin channel in RAP-treated parasites appears over-saturated. Please adjust intensity to accurate representation.

4) Extended Data Fig. 4e: The control and PfARK1-KO schizonts appear at different developmental stages (control: early/mid schizonts; RAP-treated: late schizonts). Please clarify whether these images were acquired at comparable time points, as stage differences may confound phenotype interpretation.

5) Fig. 3f: Please provide higher-magnification merged images showing GAP45, tubulin, SYTOX, and NHS-ester channels to better illustrate the defective merozoite phenotype in RAP-treated cells. The RAP-treated schizonts shown appear at an early segmentation stage, indicated by partially formed GAP45 signals and short subpellicular microtubules. In contrast, control schizonts are fully formed GAP45 sheaths surrounding individual merozoites with well-developed subpellicular microtubules. Because subpellicular microtubule length varies with stage, these images cannot be directly compared to support the statement that "subpellicular MT were either not observed in the cell or were mislocalised" (line 233 to 234). Please include representative fully segmented schizonts or clarify this limitation.

6) Line 325 and Fig. 5dJ: The statement that "MTOCs that appeared detached from the nucleus in the ark1 PTD line" is not clearly supported. In Fig. 5dJ, the inner MTOC remains within the nucleus, and Extended Data Fig. 6bA, B and D, the inner MTOC is not evident. More convincing or annotated images are needed.

Minor comments:

1) Line 84: The localization of PfARK1-3 in gametocyte was previously reported (PMID: 39235260), showing PfARK1 is specific to male gametocytes.

2) Figure 1b right panel: The localization of PfARK1-3 was previously described in PMID: 39235260 and PMID: 21166904.

3) Line 114-115: The statement that "ARK1 is expressed in trophozoite and schizont stages, but not detectable in late schizonts and all developing merozoites (Fig. 2a)" is not fully supported. In Fig. 2a, the "trophozoite" image resembles an early schizonts (two nuclei), and the "late schizont" image resembles a segmented schizont. PfARK1 remains visible in late schizonts (Fig. 2b second panel) and was previously shown to persist (PMID: 39235260).

4) Line 153-155, Fig. 2e: The data suggest that PbARK1 is not localized to spindle poles but instead associate with axonemal structures, as indicated by co-localization with α -tubulin. Please provide more representative images to support this interpretation.

- 5) Extended Data Fig 2d: The “late schizont” image actually presents a segmented schizont. Additionally, strong background fluorescence in Extended Data Fig. 2d-e obscures PbARK1 foci. Highlighting these regions would improve clarity.
- 6) Extended Data Fig. 2f: The localization of PbARK1 relative to the mitotic spindle or spindle poles remains ambiguous. Higher-quality representative images are needed.
- 7) Line 165 and Fig. 2f-g right panels: In Fig. 2f right panel, the PbARK1-GFP signal is broadly distributed within the nucleoplasm and does not clearly associate with the spindle or PbNDC80-mCherry. Similarly, in line 169 and Fig. 2g right panel, PbARK1-GFP is not distinctly localized to the spindle. Additional higher-resolution images or z-sections are recommended.
- 8) Extended Data Fig. 6a: The highlighted squares do not match the zoom-in regions.
- 9) Line 417-418: The claim that “Its colocalisation with kinetochore NDC80 and ARK2 markers further supports a dual role in spindle organization, kinetochore segregation, and spindle pole integrity” is not directly demonstrated. Consider revising to “potentially supports” rather than asserting it as proven.
- 10) Line 427: The statement “In *P. falciparum*, ARK1 is required for proper spindle elongation” is unclear. Long spindle microtubules resembling hemispindles remain visible in RAP-treated parasites (Fig. 3d, third panel), suggesting elongation is not fully abolished. Please clarify or provide quantitative evidence.

Reviewer #3

(Remarks to the Author)

Summary: This study comprehensively defines the role of a key protein called ARK1 during mitosis of malaria parasites both in the asexual blood-stage, and during male gamete formation. Mitosis is perhaps the most well studied process in all of biology, but by contrast our understanding of mitosis and the proteins that control it in malaria parasites is relatively poor. The role of ARK1 is explored using a range of microscopy modalities (live-cell, TEM, expansion microscopy), transcriptomics, pull-downs, and an evolutionary/bioinformatic analysis. This study represents one of the more comprehensive protein-characterisation projects in malaria parasites and contributes to a growing body of literature on aurora-like kinases in apicomplexan parasites. These findings will definitely be of interest to the parasite cell biology community, and the evolutionary biology findings may be useful to the broader study of mitotic kinases. While not discussed extensively by the authors, the aurora-like kinases do represent attractive targets for the development of novel antimalarials, and so findings from this study may also be useful in that context.

Overall the experiments in this study are sound, and in my opinion could likely be published without the addition of new experiments. The way in which some of the data has been presented, or referred to, however could be improved. Below I have provided a number of comments that I have deemed as major (essential to address), minor (would be good to address), and some comments on spelling, syntax etc. Even though there are many comments, it is likely that these could be completely or mostly addressed in a reasonably short time-frame with changes to the text, figure presentation, or data analysis.

Major Comments

- Throughout the text a range of terms for different biological processes are conflated with each other. Individually, these are not major problems, but collectively they make it confusing to determine the interpretation of some of the experiments. These should all be easily fixable with text changes. Below I will provide a few examples of this:
 - o Line 49 describes schizogony as a form of mitosis, when schizogony refers to the mode of cell division
 - o Lines 124 – 125: refers to parasites undergoing mitotic division, when the data regards nuclei undergoing karyokinesis
 - o Line 205 mentions looking at asexual cell division when karyokinesis is assessed
 - o Line 393, the first line of the discussion describes ARK1 as a central organiser of mitosis, but the term mitosis is not used at all when describing the phenotype of ARK1
 - o Line 399 implies ARK1 has a phenotype upstream of chromosome segregation. I assume this is meant to be chromatid segregation, but in either case, this is not directly assessed in any experiment.
 - o Lines 487-490 mention an atypical mechanism of cell division when it seems to be discussing mitosis.
- The statistical analyses performed in Figures 3a, Extended data Figure 1f, 4a, and 4b are not appropriate. Because the control is normalised to 1, its standard deviation = 0. I don't think this impacts the interpretation of the results hugely, but they should either be presented without this statistical test, or be presented without normalisation with a statistical test.
- The interpretation of ARK1 localisation seems inconsistent across the manuscript. In the *P. falciparum* imaging, it was determined that ARK1 was only present alongside a mitotic spindle with no significant fluorescence outside this. Based on this, it seems reasonable that the diffuse cytosolic expression seen in the *P. berghei*-ARK1-GFP parasites is likely background. In some lifecycle stages like oocysts, sporozoites, and liver-stages this diffuse cytoplasmic staining is inferred as representing the localisation of ARK1, which in asexual and activated gametocytes it is ignored in favour of the punctate staining. The authors should be clear on when and why they think the observed fluorescence represents true ARK1 localisation vs off-target/background signal. Also, the observation of ARK1 in the Golgi of salivary gland sporozoites should be discussed.

- The authors conclude that rhoptries were not formed properly or were absent following ARK1 iKO. The images in Figure 3F don't seem to support this conclusion, as almost every merozoite has a visibly recognisable rhoptry pair.
- I feel that the transcriptomic results are slightly over-interpreted. The only consistently downregulated group of proteins are ribosome-related, and none of them are mentioned individually (only as part of the GO analysis). The authors interpret the lack of ribosome biogenesis as indicative of cell cycle arrest and translational inhibition, but an alternative interpretation is that immediately post-fertilisation the parasites prepare to make new ribosomes.

Minor Comments

- The abstract says Plasmodium lacks a Bub1 homologue, but the Bub1 orthogroup on OrthoMCL contains the *P. falciparum* protein Pf3D7_1112100.
- Lines 59-61 describes gametogeny as atypical because of an MTOC that spans the nuclear envelope and makes microtubules on both sides. With the exception of making an axoneme, this is also what happens in both asexual blood-stage parasites and during gametocytogenesis.
- The naming of the centriolar plaque are very confusing across the manuscript. The structure is variously referred to as the MTOC (generally), the inner and outer MTOC, the centrosome, centriolar plaque, and spindle pole body. My suggestion for the least ambiguous way to define this structure is to call the overall structure the centriolar plaque, and specifically refer to the inner (intranuclear) and outer (cytoplasmic) regions of it. In any case, the naming conventions should be harmonised across the manuscript.
- In the introduction it should also be stated that the Wyss et al study (ref 17) determined the localisation of ARK1 in *P. falciparum* gametocytes
- Line 142 should specify activated gametocytes
- It seems likely that impaired centriolar plaque segregation is the earliest occurring phenotype observed following ARK1 KD/iKO. All the other phenotypes, like impaired cytokinesis, karyokinesis, etc. are therefore most likely to be downstream consequences of impaired centriolar plaque segregation. It isn't clear in the text that these are likely a related series of events.
- Did the authors check to see if there is a compensatory ARK2 upregulation in the ark1PTD cell line, like there is following ARK1 iKO?
- Line 310 says no exflagellation was observed, while line 333 says exflagellation was rarely observed.
- Line 434-435 discusses a failure in MTOC-basal body segregation. Previous studies of inner centriolar plaque proteins have seen separation of the inner and outer centriolar plaque following knockdown. This seems similar enough to me to be worth mentioning here.
- Line 481 says ARK1 and ARK2 are both indispensable, but this was not shown in this study.
- Figures 2 and 3 could probably each be broken up into two figures. Figure 2 could logically be broken up into a *falciparum* and *berghei* figure. If not, the figure legends should be condensed as they're currently over a full page of text.
- For the microscopy analyses of aberrant nuclear, MSP1 and GAP45 staining, it should be better described what was defined as 'normal' and whether these were blinded.
- For Figure 1a and 2a: The outer centriolar plaque is attached to the parasite plasma membrane from the onset of mitosis until at least the start of segmentation, but the diagram implies they float freely in the cytoplasm
- Figure 1a says two hemispindles but depicts an interpolar spindle
- In Figure 1b it says cytokinesis (central spindle), but this structure doesn't exist when the parasite is doing cytokinesis. Also the diagram includes astral microtubules, which we don't have evidence of in Plasmodium.
- I don't think it is fair to say ARK1-3 are absent in *Cryptosporidium* there are 6 *C. parvum* proteins in the ARK Orthogroup (Cdg1_1220, Cgd1_2630, Cgd8_750, Cgd3_3040, Cgd2_1830, Cgd6_5240). Comparing these proteins against the consensus sequences in Figure 6, it seems likely that Cgd3_3040 is the *C. parvum* homologue of ARK2.
- Considering the strength and frequency of all the phenotypes quantified in Figure 3, it seems surprising that there is a <50% growth defect at this point in time. I think this is worth discussing in more detail, and also mentioning whether the reasonably modest growth defect might be due to the compensatory upregulation of ARK2.
- In Extended Data Figure 2d, the Hoechst staining seems to be almost fully excluded from the forming merozoites. Is there a channel alignment issue or something there?
- The expression profiles of ARK1 and AMA1 are very different. Did the authors determine if the ark1-PTD parasites have approximately normal blood-stage growth? This is important for interpreting the results of the bite-back experiment. Also, cause AMA1 is expressed in sporozoites, would it be expected to get a second wave of ARK1 expression during late oocyst development?
- If I am interpreting Extended data Figure 5 correctly, roughly 50% of sporozoites produced sporozoites make it to the salivary gland in both WT and ark1-PTD parasites. From this, can it be inferred that the sporozoites that get produced are likely still invasive?
- Figure 4e is missing some scale bars and the ark1PTD oocysts take up less of the FOV than the WT, making them look much smaller than the quantification would indicate.
- Figure 3f, DMSO Tubulin is hugely over-exposed
- Figure 2h the Hoechst looks very strange in the 3D-SIM image, is this expected?

Spelling, grammar, syntax, etc.

- Lines 57 and 63: De novo is not italicised.
- Line 125 should say nuclei, or parasite nuclei, instead of parasites
- Line 140: I think "along microtubules" rather than "at microtubules" is probably a better description of the localisation
- Line 275: 'bite-back experiments' should be explained in more detail
- Line 312-313: "led us to examine whether this" can be deleted

- Line 339-340: I don't understand "to confirm the logic of the observed subcellular localisations".
- Line 353-358 is more discussion rather than results
- Extended data Figure 2c: What do the two annotated sizes correspond to?

Version 1:

Reviewer comments:

Reviewer #1

(Remarks to the Author)

The authors have adequately addressed my previous concerns. This is a high-quality study that provides new insights into the MTOC of Plasmodium parasites.

Reviewer #2

(Remarks to the Author)

The authors have addressed most comments in this revised version, and the clarity of the manuscript has improved. To further strengthen data presentation, as noted in my previous major comments 2, 3 and 5, I encourage the inclusion of zoomed-in views of individual mitotic spindle structures in Fig. 2b and zoomed-in views of the highlighted phenotypic regions in Fig. 4d and 4f of *P. falciparum*, similar to Fig. 5g for *P. berghei*. Without such magnification, several described phenotypes-such as "centrin aggregates were seen away from the nuclei and disorganised" (line 229) in Fig. 4d and "subpellicular MT were either not observed or were mislocalised" (lines 249-250) in Fig. 4f- are difficult to discern. I was also unable to locate Extended Data Fig. 3d-e referenced in the response to minor comment 5.

Reviewer #3

(Remarks to the Author)

The authors have done a very thorough job responding to all of my comments, and these changes have no impact on the significance/novelty of the findings of this study. In my opinion this study is now ready for publication.

My only unaddressed comment is regarding the statistical analysis in Figure 4a. Here the authors use ANOVA to calculate what they're presenting as a pairwise analysis (which should be a t-test or similar instead). In either case, statistical tests where the algorithm includes sample variance (like ANOVA or t-tests) cannot be performed on this data because the variance of the DMSO treated sample = 0 due to normalisation. This means the statistical significance of these comparison gets inflated because true sample variances are not being compared.

Response to reviewers

We thank all the reviewers for their comprehensive evaluation of our manuscript and constructive feedback.

We have performed additional experiments and further quantification, revised the manuscript's text and figures to improve clarity and accuracy, and addressed most comments of the reviewers. Below is a point-by-point response to the reviewers' comments.

As suggested by the reviewers, the original Figure 2 is now divided into two separate figures (Figure 2 for *P. falciparum* results and 3 for *P. berghei* results, respectively). **Please note that the figure numbers mentioned below refer to this revised numbering. The line numbers mentioned below refer to those in the clean version of the manuscript.**

Reviewer comments

Reviewer #1 (Remarks to the Author):

The manuscript by Nagar and colleagues evaluates the function of ARK1 in Plasmodium parasites. Using the rodent malaria, Plasmodium berghei, and the human Plasmodium falciparum, ARK1 is hypothesized to be part of the inner MTOC and to influence spindle/kinetochore dynamics. The findings are most striking during male gamete formation.

The manuscript combines elegant genetic, cell biology, and evolutionary biology to understand the function of the divergent ARK1 in Plasmodium. This is a nice study that combines findings in two Plasmodium species. The findings are largely well-supported. I have suggested some additions and changes that I think will improve the study.

Major

1. Claims about the timing of ARK1 expression in line 124-125 are likely true but are only weakly supported by the provided data of two schizonts. It would be helpful for the reader to have some quantification of this phenotype. One could imagine counting the number of nuclei in each stage of their karyokinesis cycle and noting if ARK1 is present or absent. I do not doubt the result shown, but it would be stronger with quantification. Line 127-137, please explain why the localization of PbARK1 is diffuse when this localization is not mentioned or observed for PfARK1. This is likely due to method differences, but the claim that the localization is the similar is not fully supported without further explanation.

The quantification of parasites with ARK1 staining has been added to Fig. 2c.

Methodological differences and different reporters (GFP for *P. berghei* and HA for *P. falciparum*) probably are the reason. Live cell imaging of GFP was done to localize PbGFP-ARK1, whereas indirect immunofluorescence (IFA) with fixed cells was used to locate HA-PfARK1. These two methodologies may have different sensitivities and IFA may give some cytoplasmic autofluorescence.

2. The RNAseq paragraph (lines 279-300) does not seem to add any useful information. I think this is distracting and detracts from the more interesting story that is unfolding. This should be removed because the observations/hypotheses are not experimentally tested.

We have removed this paragraph from the main text as suggested, and relocated it to the supplementary note. A brief paragraph describing the RNA-seq analysis is now included (Lines 301-306).

3. The use of immunoprecipitation to identify INCENP1 and INCENP2 is a strong piece of data. It would be nice, if possible, to show colocalization of ARK1 and these proteins. While it seems extremely likely that they will colocalize, it is good technique to validate findings from mass spectrometry with additional experiments. Because these structures are stable and associated with microtubules, they may “interact” or immunoprecipitated over large distances.

We strongly agree with the reviewer on the importance of validating mass spectrometry interactions with colocalization studies to rule out indirect associations. In fact, a preprint by Roques et al. (2025)¹, published subsequent to our submission, explicitly demonstrates that PbARK1 co-localises with INCENP1 and INCENP2 at spindle poles, spindles, and kinetochores in *Plasmodium berghei*. These independent imaging studies provide the requested visual confirmation of co-localisation, strongly supporting our immunoprecipitation results. We have opted to discuss the findings of Roques et al. in the main text to corroborate our interaction data (Lines 356-358), rather than repeating these specific experiments with additional mice and thereby adhering to the ethical principle of minimising animal usage.

Correction of terminology (INCENP2 → INCENP-A; INCENP1 → INCENP-B): In our initial submission, we referred to the two Plasmodium INCENP proteins as INCENP1 and INCENP2, following the nomenclature from studies on *Toxoplasma gondii*, where two INCENP paralogs had been identified. We hypothesised that these were direct orthologs between Plasmodium and Toxoplasma.

However, further phylogenetic inspection, in the light of the recent Roques et al independent study, suggested that a duplication giving rise to two INCENP paralogs likely occurred independently in different apicomplexan lineages (i.e. there is no clear evidence for an INCENP duplication in the shared ancestor of Coccidia and Hematozoa). Thus, the two Plasmodium INCENPs are not orthologs of the Toxoplasma INCENP1/2 pair. Most importantly, both our dataset and that of Roques et al. show that the Plasmodium INCENP originally called *INCENP2* in our manuscript (INCENP1 in Roques et al.) is the constitutive ARK1-interacting paralog, whereas the protein originally named *INCENP1* interacts with ARK1 only during meiosis. To avoid confusion and incorrect evolutionary inference, we have renamed Plasmodium INCENP2 as INCENP-A and Plasmodium INCENP1 as INCENP-B throughout the manuscript. This naming reflects functional roles rather than unsupported assumptions of

orthology. All text, figures, legends, and Supplementary Tables have been updated accordingly (Lines 346-409, 486-503, Figure 7, Extended Data Fig. 8 and 9, and Supplementary Table 5).

Minor

1. Figure 1b is a nice figure to give the reader a map of the proposed localization and function. The small blue “nucleoplasm” dot is a bit odd and was confusing for me. I think that this would be better to move out of the center of the panel and into a legend box or something. This extra dot seems like it should be a structure when it is not really anything.

We thank the reviewer for pointing out this confusion. In the revised Figure 1b, we have removed the “nucleoplasm” dot that could be interpreted as a biological structure, and redesigned the panel to portray more clearly the locations of various Aurora kinases.

2. Line 123 – the term “polar ends” should be clearly defined.

We have replaced the term "polar ends" with "growing mitotic spindles" for improved clarity.

3. Fig 2E – the GFP antibody staining has high background. Can a different cell be selected? Perhaps a different gel preparation and a different antibody preparation (or source of antibody) will help to reduce this unfortunate background.

We have prepared new gels and re-acquired the data using a different anti-GFP antibody. As a result, the background was reduced, and we saw a clearer localization of ARK1-GFP to the spindle and spindle poles (Figure 3c).

4. Line 197 – the statement “ a significant number” needs to be backed up by a statistical measurement or removed/changed.

Quantification and statistics are now provided in revised Extended Data Fig. 5c to support this statement.

5. Line 234-236 – The primary defect is abnormal karyokinesis. The cytokinesis defect is likely entirely secondary to this primary defect. It should be stated very explicitly that ARK1 has little to do with cytokinesis. While this is implied, for the non-expert reader it should be stated clearly.

We agree with the reviewer that this is a critical distinction, and that the observed cytokinesis defects are most likely the consequences of the primary failure in karyokinesis. To ensure that this distinction is evident we have revised the text, stating explicitly that the impairment in cytokinesis is likely a downstream effect of a primary defect in nuclear division, rather than the result of a direct role of ARK1 in cytokinesis (line 251-253).

6. In my opinion, the fig 6b could be moved to a supplement. The AlphaFold3 server is publicly available and anyone could generate these models on their own.

We appreciate the reviewer's suggestion; however we would prefer to retain this figure in the main paper. Our argument is that although AlphaFold3 is publicly accessible, the specific co-fold predictions of ARK1 with INCENP-A/B structure presented in Fig. 6b required custom modelling tailored to this study, and are not available elsewhere. Although single-protein AlphaFold2 predictions are widely accessible, these interaction-focused AF3 models provide novel structural insight into the apicomplexan CPC architecture, which directly supports our suggestions about conserved IN-box-kinase interfaces and the divergent CxC motifs. These structural features are integral to our mechanistic interpretation and are discussed explicitly in the main text, therefore we feel that they should be retained in the primary figure to improve clarity and strengthen the narrative.

Reviewer #2 (Remarks to the Author):

Summary:

In this manuscript, Nagar and colleagues present a comprehensive investigation of Aurora-related kinase 1 in *P. falciparum* schizogony and *P. berghei* male gametogony, combining live-cell imaging, expansion microscopy, and electron microscopy. The authors demonstrate that PbARK1 is essential for spindle formation, MTOC separation, exflagellation, and parasite transmission, while PfARK1 disruption impairs spindle assembly and normal karyokinesis/cytokinesis. Proteomics analysis suggests that PbARK1 interacts with INCENP1/2 to form a chromosome passenger complex.

Overall, the study provides valuable insight into ARK1 function in both asexual and sexual stages of Plasmodium, contributing to a better understanding of how ARK1 regulates cell division through distinct mitotic mechanisms in schizogony and gametogony. However, several methodological and data presentation issues need to be addressed. Specially, the internal HA-tag insertion in PfARK1 requires further justification. Additionally, the PfAK1 expansion microscopy images lack sufficient resolution or stage comparability to convincingly support claims that PfARK1 deficiency causes centriolar plaque mis-segregation or separation from the nucleus, as well as altered subpellicular microtubule polymerization or localization. Therefore, additional clarification and higher-resolution data are required before publication.

Major comments:

1) Line 112 and Extended Data Fig. 1a: The authors describing generating a HA-PfARK1-loxP parasite line, which appears to be N-terminally HA-tagged. However, Extended Data Fig. 1a shows that the HA tag was inserted internally within the PfARK1 coding sequence rather than at the N- or C-terminus. Although the localization pattern resembles PfARK1-GFP reported

previously, and this line was used for conditional PfARK1 disruption, it remains unclear whether this internal tagging affects the protein's native function or folding. Please clarify the rationale for this strategy and specify whether the HA tag lies in an unstructured or non-conserved region.

We hereby confirm that the HA-tag was introduced at the N-terminus of ARK1 as indicated in the schematic in Extended Fig. 1a. A small homology arm from the 5'-region of the PfARK1 gene was used for recombination to insert a loxP site, followed by a sequence for a 2A skip peptide spanning γ DHODH, and downstream a recodonized sequence coding for full-length ARK1 with an HA-tag at its N-terminus and another loxP site was introduced. Following recombination, N-terminal HA-tagged ARK1 was expressed, and following rapamycin-mediated excision, this gene sequence was deleted. A similar Selection-Linked Integration (SLI) approach has been used in the past by us² and others³. We have modified the figure legend to clarify the construction and the methods section also has these details (Lines 551-560 and Extended Data Fig 1).

2) Lines 113-125 and Fig. 2a-b: PfARK1 localization has been reported previously (PMID: 21166904; PMID: 39235260). The data here are confirmatory and should be stated as consistent with those earlier findings.

For Fig. 2a-b, please provide higher-magnification PfARK1/Tubulin/NHS-ester merged images to improve clarity and data presentation.

We include now a magnified PfARK1/Tubulin/NHS-ester image and cite the mentioned papers.

3) Fig. 3d: Please provide higher-magnification merged images combining the tubulin, centrin, SYTOX, and NHS-ester channels to better illustrate the described phenotype. Current overview images lack resolution to convincingly support the statement that "the nuclei appeared unsegregated with associated multiple centrin foci, and centrin aggregates were seen away from the nuclei and disorganized" (line 211-212). In addition, the tubulin channel in RAP-treated parasites appears over-saturated. Please adjust intensity to accurate representation.

We include now the magnified tubulin/centrin/ SYTOX/NHS-ester image, and a new image panel (revised Figure 4d) that displays this defect better has been included. These images, together with additional images in Extended Fig 5e, collectively show the presence of multiple centrin foci associated with nuclear "aggregates". In RAP-treated parasites spindles are not formed and residual tubulin is visualised as random "aggregates or speckles". This distribution gives an impression of being over-saturated; however as suggested by the reviewer we have adjusted the brightness and replaced one panel of the figure (revised Figure 4d).

4) Extended Data Fig. 4e: The control and PfARK1-KO schizonts appear at different developmental stages (control: early/mid schizonts; RAP-treated: late schizonts). Please clarify

whether these images were acquired at comparable time points, as stage differences may confound phenotype interpretation.

The experiments were performed simultaneously with control and PfARK1-KO schizonts, however the growth of the KO parasites was impaired and the stage of schizont development was skewed. Therefore, for better comparison we now include additional images of late schizonts for control parasites and early schizonts for RAP-treated parasites. It is clear from these data that in late schizonts, which have undergone more rounds of nuclear division and CP duplication, the defects due to lack of nuclear division are even more apparent as the number of centrin foci is much higher.

5) Fig. 3f: Please provide higher-magnification merged images showing GAP45, tubulin, SYTOX, and NHS-ester channels to better illustrate the defective merozoite phenotype in RAP-treated cells. The RAP-treated schizonts shown appear at an early segmentation stage, indicated by partially formed GAP45 signals and short subpellicular microtubules. In contrast, control schizonts are fully formed GAP45 sheaths surrounding individual merozoites with well-developed subpellicular microtubules. Because subpellicular microtubule length varies with stage, these images cannot be directly compared to support the statement that “subpellicular MT were either not observed in the cell or were mislocalised” (line 233 to 234). Please include representative fully segmented schizonts or clarify this limitation.

We now include a merged image of GAP45/tubulin/ SYTOX/NHS-ester channels (revised Fig. 4f). In PfARK1-depleted parasites, proper schizont segmentation was largely not observed, and these parasites "arrested" prior to segmentation. These parasites did not undergo proper, complete segmentation, which either caused or was a consequence of disrupted SMTs. This point is now clarified in the manuscript (Lines 243-253). Some additional images of parasites that showed some segmentation were selected and these are included in Extended Fig. 5f.

6) Line 325 and Fig. 5dJ: The statement that “MTOCs that appeared detached from the nucleus in the *ark1* PTD line” is not clearly supported. In Fig. 5dJ, the inner MTOC remains within the nucleus, and Extended Data Fig. 6bA, B and D, the inner MTOC is not evident. More convincing or annotated images are needed.

We apologise for the confusion caused by an incomplete description of the images. The intention had been to demonstrate that some basal body MTOCs appear detached from the nucleus. We have now included additional annotation ("dBB") to indicate basal body MTOCs that appear to be detached and provided an additional image (Extended Data Fig. 7bC), although since TEM provides only two-dimensional images, it does not prove definitively that these basal body MTOCs are truly detached from the nucleus. Using 3D optical sectioning of *ark1PTD* gametocytes we have clearly seen that some basal body MTOCs are located away from the nucleus (Figure 5g and Extended Data Fig. 7a), and these TEM observations provide supporting evidence for this observation.

Minor comments:

1) Line 84: The localization of PfARK1-3 in gametocyte was previously reported (PMID: 39235260), showing PfARK1 is specific to male gametocytes.

We have revised the text and Figure 1b to clarify the known locations and cite previous work^{4, 5}.

2) Figure 1b right panel: The localization of PfARK1-3 was previously described in PMID: 39235260 and PMID: 21166904.

The subcellular locations of ARK1-3 determined previously are shown in Figure 1b. The italicised text refers to putative complexes that are known to scaffold specific aurora kinases. To avoid confusion, we have attempted to clarify this point.

3) Line 114-115: The statement that “ARK1 is expressed in trophozoite and schizont stages, but not detectable in late schizonts and all developing merozoites (Fig. 2a)” is not fully supported. In Fig. 2a, the “trophozoite” image resembles an early schizonts (two nuclei), and the “late schizont” image resembles a segmented schizont. PfARK1 remains visible in late schizonts (Fig. 2b second panel) and was previously shown to persist (PMID: 39235260).

We agree that the image is of an early schizont; we have corrected the figure and its labelling.

4) Line 153-155, Fig. 2e: The data suggest that PbARK1 is not localized to spindle poles but instead associate with axonemal structures, as indicated by co-localization with α -tubulin. Please provide more representative images to support this interpretation.

We have prepared new expansion gels and acquired new data, which are incorporated in Figure 3. These images, including NHS-ester and α -tubulin staining, show more clearly that PbARK1 is located in particular at the spindle poles and not axonemal structures (white arrows in Figure 3c).

5) Extended Data Fig 2d: The “late schizont” image actually presents a segmented schizont. Additionally, strong background fluorescence in Extended Data Fig. 2d-e obscures PbAKR1 foci. Highlighting these regions would improve clarity.

We agree that this stage is a segmented schizont. We have revised the label (“segmentor”) and replaced the image with a more representative one. We also highlight the PbARK1 foci in Extended Data Fig. 3d-e to improve clarity.

6) Extended Data Fig. 2f: The localization of PbARK1 relative to the mitotic spindle or spindle poles remains ambiguous. Higher-quality representative images are needed.

These data were from an IFA study, and the location of PbARK1-GFP was not as clearly demonstrated as in live-cell or U-ExM imaging studies. Therefore, we have decided to remove these IFA data, and we have revised the manuscript to refer instead to the U-ExM data.

7) Line 165 and Fig. 2f-g right panels: In Fig. 2f right panel, the PbARK1-GFP signal is broadly distributed within the nucleoplasm and does not clearly associate with the spindle or PbNDC80-mCherry. Similarly, in line 169 and Fig. 2g right panel, PbARK1-GFP is not distinctly localized to the spindle. Additional higher-resolution images or z-sections are recommended.

We have re-acquired these data and added additional higher-resolution SIM and U-ExM images to the main and supplementary figures that show more clearly the co-localisation of PbARK1-GFP with PbNDC80-mCherry and PbARK2-mCherry (Figure 3e, g and Extended Data Fig. 3b, c). We have also moved to the main figures the live-cell imaging data that show the co-localisation in activated gametocytes. As described above, the original Figure 2 has been divided into two separate figures (revised Figures 2 and 3) to distinguish the *P. falciparum* results from those of *P. berghei*.

8) Extended Data Fig. 6a: The highlighted squares do not match the zoom-in regions.

We have corrected the positions of the highlighted squares (revised Extended Data Fig 7a).

9) Line 417-418: The claim that “Its colocalisation with kinetochore NDC80 and ARK2 markers further supports a dual role in spindle organization, kinetochore segregation, and spindle pole integrity” is not directly demonstrated. Consider revising to “potentially supports” rather than asserting it as proven.

As suggested by the reviewer, we have revised this sentence and use “potentially supports” to reflect a more cautious interpretation (Lines 437-440).

10) Line 427: The statement “In *P. falciparum*, ARK1 is required for proper spindle elongation” is unclear. Long spindle microtubules resembling hemispindles remain visible in RAP-treated parasites (Fig. 3d, third panel), suggesting elongation is not fully abolished. Please clarify or provide quantitative evidence.

We agree that there is residual staining of tubulin, which we have referred to as “speckled tubulin staining” in the text (Lines 225-228). Although RAP-treated parasites show a loss of spindles, a few parasites still exhibit aberrant or speckled tubulin staining. For clarity, we have replaced the parasite image in the revised Fig. 4d (last two panels), which represent these

defects with more clarity. It is clear from Fig. 3d (now Figure 4 and Extended Data Fig. 5e that RAP-treated parasites largely lack proper spindles.

Reviewer #3 (Remarks to the Author):

Summary: This study comprehensively defines the role of a key protein called ARK1 during mitosis of malaria parasites both in the asexual blood-stage, and during male gamete formation. Mitosis is perhaps the most well studied process in all of biology, but by contrast our understanding of mitosis and the proteins that control it in malaria parasites is relatively poor. The role of ARK1 is explored using a range of microscopy modalities (live-cell, TEM, expansion microscopy), transcriptomics, pull-downs, and an evolutionary/bioinformatic analysis. This study represents one of the more comprehensive protein-characterisation projects in malaria parasites and contributes to a growing body of literature on aurora-like kinases in apicomplexan parasites. These findings will definitely be of interest to the parasite cell biology community, and the evolutionary biology findings may be useful to the broader study of mitotic kinases. While not discussed extensively by the authors, the aurora-like kinases do represent attractive targets for the development of novel antimalarials, and so findings from this study may also be useful in that context.

Overall the experiments in this study are sound, and in my opinion could likely be published without the addition of new experiments. The way in which some of the data has been presented, or referred to, however could be improved. Below I have provided a number of comments that I have deemed as major (essential to address), minor (would be good to address), and some comments on spelling, syntax etc. Even though there are many comments, it is likely that these could be completely or mostly addressed in a reasonably short time-frame with changes to the text, figure presentation, or data analysis.

Major Comments

- Throughout the text a range of terms for different biological processes are conflated with each other. Individually, these are not major problems, but collectively they make it confusing to determine the interpretation of some of the experiments. These should all be easily fixable with text changes. Below I will provide a few examples of this:

- o Line 49 describes schizogony as a form of mitosis, when schizogony refers to the mode of cell division

We agree with the reviewer's distinction between mitosis and cell division, and that schizogony refers to a mode of cell division rather than a form of mitosis. We have revised the sentence to clarify this point (Lines 52-54).

- o Lines 124 – 125: refers to parasites undergoing mitotic division, when the data regards nuclei undergoing karyokinesis

In these experiments, PfARK1 localisation was assessed during parasite division. Since it was only localised specifically in parasites that possessed mitotic spindles, we mention that its expression was "only when parasites undergo mitotic division" (Lines 136-137).

o Line 205 mentions looking at asexual cell division when karyokinesis is assessed

The statement now reads "Next, efforts were made to investigate if PfARK1 regulates MTOC duplication and spindle formation during asexual division of the parasite" (Lines 221-222).

o Line 393, the first line of the discussion describes ARK1 as a central organiser of mitosis, but the term mitosis is not used at all when describing the phenotype of ARK1

While we described specific phenotypic defects such as impaired spindle formation, MTOC separation, and karyokinesis, we did not explicitly categorise these as defects in "mitosis" within the Results section. To address this point and to support more fully our conclusions in the Discussion, we have revised the Results section to explicitly state that ARK1 depletion leads to defects in mitosis (Lines 230-232, 343-344).

o Line 399 implies ARK1 has a phenotype upstream of chromosome segregation. I assume this is meant to be chromatid segregation, but in either case, this is not directly assessed in any experiment.

We agree that "sister chromatid segregation" is the appropriate terminology. We also acknowledge that while our data show that ARK1 depletion leads to failures in spindle formation and MTOC separation—thereby preventing segregation—we did not strictly assess its hierarchical position upstream of the segregation machinery itself. We have revised the text to replace "chromosome segregation" with "sister chromatid segregation" and to state more accurately that ARK1 is essential for the spindle and MTOC functions required for these processes (Line 418-420).

o Lines 487-490 mention an atypical mechanism of cell division when it seems to be discussing mitosis.

We agree that given the focus of the work on the molecular machinery regulating chromosome segregation and the divergence of canonical regulators like the CPC, the term "mitosis" is more appropriate than the broader term "cell division". We have revised the text accordingly (Lines 526-529).

- The statistical analyses performed in Figures 3a, Extended data Figure 1f, 4a, and 4b are not appropriate. Because the control is normalised to 1, its standard deviation = 0. I don't think this

impacts the interpretation of the results hugely, but they should either be presented without this statistical test, or be presented without normalisation with a statistical test.

To clarify the statistical analyses in these figures (in the revised Figure 4a and Extended data Figure 1f, 4a, and 4b), the purpose of the quantitation was to compare change in parasitemia in RAP-treated and DMSO-treated (control) parasites at the time points indicated in three replicate experiments. For this purpose, the control parasitemia at each time point was treated as 1 and the fold difference in parasitemia was calculated for RAP-treated parasites. For instance, if the control parasitemia was 4 % and the RAP-treated parasitemia was 2%, then the fold difference would be 0.5, and at the following time point if it was 7% versus 2% the fold difference would be 0.28. We think that the statistical analysis of these data for 3 replicates provides an appropriate comparison, and we and others have used this method in previous studies⁶.

- The interpretation of ARK1 localisation seems inconsistent across the manuscript. In the *P. falciparum* imaging, it was determined that ARK1 was only present alongside a mitotic spindle with no significant fluorescence outside this. Based on this, it seems reasonable that the diffuse cytosolic expression seen in the *P. berghei*-ARK1-GFP parasites is likely background. In some lifecycle stages like oocysts, sporozoites, and liver-stages this diffuse cytoplasmic staining is inferred as representing the localisation of ARK1, which in asexual and activated gametocytes it is ignored in favour of the punctate staining. The authors should be clear on when and why they think the observed fluorescence represents true ARK1 localisation vs off-target/background signal. Also, the observation of ARK1 in the Golgi of salivary gland sporozoites should be discussed.

We attribute the differences in "diffuse" signal primarily to the imaging methodology used: *P. falciparum* parasites were analysed by IFA on fixed cells, where it is possible that soluble cytosolic protein may be washed away, whereas *P. berghei* parasites were analysed using live-cell GFP imaging, which detects all the GFP-tagged protein within the cell.

We agree that the distinct focal staining at the spindle represents the location of active ARK1 during mitosis. We interpret the diffuse signal observed in several life cycle stages of *P. berghei* as a cytosolic pool of the protein, rather than non-specific background, although we cannot rule out some level of cellular autofluorescence or background. To ensure consistency and clarity throughout the manuscript, we have revised the text for oocyst, sporozoite, and liver stages to describe the signal as a "diffuse cytosolic distribution" without inferring a specific location. For the signal in salivary gland sporozoites, we use the same description, and have refrained from speculating on a Golgi location since we lack specific Golgi markers.

- The authors conclude that rhoptries were not formed properly or were absent following ARK1 iKO. The images in Figure 3F don't seem to support this conclusion, as almost every merozoite has a visibly recognisable rhoptry pair.

We agree with the reviewer that rhoptry formation was largely unaffected but their cellular location seemed to be disorganized, which may be due to aberrant cellular segmentation. We have made this correction (Lines 243-253).

- I feel that the transcriptomic results are slightly over-interpreted. The only consistently downregulated group of proteins are ribosome-related, and none of them are mentioned individually (only as part of the GO analysis). The authors interpret the lack of ribosome biogenesis as indicative of cell cycle arrest and translational inhibition, but an alternative interpretation is that immediately post-fertilisation the parasites prepare to make new ribosomes.

We agree that the transcriptomic results should be interpreted with caution and so we have moved the paragraph describing these results from the main text to the Supplementary Note. As suggested, in the revised text, we include specific examples of downregulated genes for ribosome-related proteins. In addition, we have softened the conclusion to state that a downregulation of ribosome biogenesis genes is consistent with developmental arrest, rather than claiming that it indicates stress sensing or translational inhibition.

Minor Comments

- The abstract says Plasmodium lacks a Bub1 homologue, but the Bub1 orthogroup on OrthoMCL contains the P. falciparum protein Pf3D7_1112100.

We thank the reviewer for spotting this error. Previously, we had concluded that Plasmodium lacked a Bub1 ortholog because candidate kinases lacked a clearly identifiable N-terminal TPR domain, a hallmark of the Bub1/BubR1 family in model eukaryotes. However, in a recent study using centromere-proximity proteomics combined with AlphaFold/FoldSeek analyses, a Bub1 ortholog has been identified in Plasmodium (<https://www.biorxiv.org/content/10.1101/2025.09.23.678001v1.full>), which contains a highly divergent TPR-like region. Although the sequence of this Plasmodium Bub1 is unconventional and appears to lack several motifs associated with a canonical spindle-assembly checkpoint response, its presence invalidates our original claim. We have, therefore, removed this sentence from the abstract and introduction, and now focus solely on the absence of polo-like kinases.

- Lines 59-61 describes gametogeny as atypical because of an MTOC that spans the nuclear envelope and makes microtubules on both sides. With the exception of making an axoneme, this is also what happens in both asexual blood-stage parasites and during gametocytogenesis.

We agree that an MTOC spanning the nuclear envelope is a feature shared with other stages, including asexual blood-stage parasites. Our intention was to highlight the unique feature of male gametogeny in which the MTOC connects spindle microtubules specifically with the

axoneme, a structure absent in schizogony. To avoid any confusion, we have revised the text to state explicitly that what is unique is the coordination with axoneme formation (Lines 63-68).

- The naming of the centriolar plaque are very confusing across the manuscript. The structure is variously referred to as the MTOC (generally), the inner and outer MTOC, the centrosome, centriolar plaque, and spindle pole body. My suggestion for the least ambiguous way to define this structure is to call the overall structure the centriolar plaque, and specifically refer to the inner (intranuclear) and outer (cytoplasmic) regions of it. In any case, the naming conventions should be harmonised across the manuscript.

We apologise for the confusing terminology regarding the microtubule organising centre and agree that use of terms such as "centrosome," "spindle pole body," and "centriolar plaque" interchangeably is confusing. For consistency across asexual stages and male gametogony (where basal bodies are involved), we now use a harmonised terminology throughout the manuscript. We now use the term "microtubule organising centre (MTOC)" for the overall structure and "inner MTOC" and "outer MTOC" to distinguish the intranuclear and cytoplasmic regions.

- In the introduction it should also be stated that the Wyss et al study (ref 17) determined the localisation of ARK1 in *P. falciparum* gametocytes

We have revised the Introduction to acknowledge that Wyss et al. had determined the location of ARK1 in *P. falciparum* gametocytes, and clarify that our study builds on this work to investigate the dynamics and function of ARK1 in both *P. falciparum* and *P. berghei* (Line 91-92).

- Line 142 should specify activated gametocytes

We describe the location of PbARK1-GFP both before and immediately after gametocyte activation. We have revised the text to specify clearly the activation status of the gametocytes in the relevant descriptions (Line 154).

- It seems likely that impaired centriolar plaque segregation is the earliest occurring phenotype observed following ARK1 KD/iKO. All the other phenotypes, like impaired cytokinesis, karyokinesis, etc. are therefore most likely to be downstream consequences of impaired centriolar plaque segregation. It isn't clear in the text that these are likely a related series of events.

The MTOC/CP duplication was unaltered in PfARK1-depleted parasites. However, several MTOC/CPs were found associated with unsegregated nuclei, likely due to defects in karyokinesis or nuclear segregation caused by loss of spindles (lines 230-232). We agree that

defects in karyokinesis likely impact the downstream process of cytokinesis (lines 251-253) and we now mentioned this in the Discussion as well (lines 446-450).

- Did the authors check to see if there is a compensatory ARK2 upregulation in the *ark1PTD* cell line, like there is following ARK1 iKO?

We examined the expression levels of *ark2* and *ark3* in the *ark1PTD* line using qRT-PCR. Interestingly, we found that both *ark2* and *ark3* were actually downregulated in *ark1PTD* gametocytes compared to wild type gametocytes (Figure below), which is in contrast to the compensatory upregulation observed in *P. falciparum* upon ARK1 depletion.

- Line 310 says no exflagellation was observed, while line 333 says exflagellation was rarely observed.

We thank the reviewer for pointing out this apparent inconsistency. As noted in Line 341 and Figure 5b, exflagellation occurs rarely in the total population. However, in Line 316 we refer specifically to a small subset of *ark1PTD* gametocytes which we examined by expansion microscopy, and among which we found no exflagellating cells. We have revised the text to clarify that no exflagellation was observed in the cells examined by expansion microscopy in this specific experiment, distinguishing them from the total population (Line 316).

- Line 434-435 discusses a failure in MTOC-basal body segregation. Previous studies of inner centriolar plaque proteins have seen separation of the inner and outer centriolar plaque following knockdown. This seems similar enough to me to be worth mentioning here.

We agree that the defective linkage between the inner and outer MTOC in the *ark1PTD* line resembles the phenotype reported following depletion of other inner MTOC proteins, where the coupling between inner and outer components is compromised. The Discussion has been revised to mention this similarity explicitly and we have cited the relevant literature (Lines 460-463).

- Line 481 says ARK1 and ARK2 are both indispensable, but this was not shown in this study.

The reviewer is correct, the indispensability of ARK2 was shown in previous studies on asexual blood stages^{7, 8, 9}, not in this study. To avoid confusion, the sentence has been revised to cite the prior literature for ARK2, and clarify that this study demonstrates the essential role of ARK1.

- Figures 2 and 3 could probably each be broken up into two figures. Figure 2 could logically be broken up into a falciparum and berghei figure. If not, the figure legends should be condensed as they're currently over a full page of text.

As stated above, Figure 2 has been split into two separate figures: one for *P. falciparum* and one for *P. berghei* as suggested (revised Figure 2 and 3).

- For the microscopy analyses of aberrant nuclear, MSP1 and GAP45 staining, it should be better described what was defined as 'normal' and whether these were blinded.

The revised Figure 4f legend defines normal MSP1/GAP45 staining as staining at the cell periphery, encompassing individual merozoites.

- For Figure 1a and 2a: The outer centriolar plaque is attached to the parasite plasma membrane from the onset of mitosis until at least the start of segmentation, but the diagram implies they float freely in the cytoplasm

We agree that the outer centriolar plaque (the outer MTOC) is anchored to the parasite plasma membrane during mitosis, and we have revised the diagrams in the revised Figure 1a and Figure 2a to show that the MTOC is in close proximity to the plasma membrane, correcting the previous representation where they appeared to be floating in the cytoplasm. We opted to not show a physical connection because the current image lacked the space to accommodate such detail.

- Figure 1a says two hemispindles but depicts an interpolar spindle

Recent high-resolution U-ExM studies (Liffner et al., eLife 2023) have clearly distinguished three intranuclear microtubule architectures during schizogony in Plasmodium:

- (1) Hemispindles, which are unipolar arrays extending from a single MTOC before division;
- (2) Mitotic spindles, formed after MTOC duplication and separation; and
- (3) Interpolar spindles, long microtubules connecting two fully separated MTOCs after chromatid segregation.

Hemispindles do not arise as symmetric pairs that fuse into a spindle. Instead, a single hemispindle forms, the MTOC duplicates, and then after a short delay the first bipolar spindle appears. Interpolar microtubules form later, after chromosome segregation. Thus, the formation of interpolar spindles and hemispindles is mechanistically distinct. To reflect this, we have revised Figure 1a so that it now correctly illustrates one hemispindle, followed by bipolar spindle formation and interpolar microtubules, in accordance with the data of Liffner et al. (2023) and other recent U-ExM studies.

- In Figure 1b it says cytokinesis (central spindle), but this structure doesn't exist when the parasite is doing cytokinesis. Also the diagram includes astral microtubules, which we don't have evidence of in Plasmodium.

We thank the reviewer for highlighting these inconsistencies. Our original intention in Figure 1b was to illustrate the range of Aurora kinase locations known from model eukaryotes, in which open mitosis includes astral microtubules and a well-defined central spindle during cytokinesis. We agree that this approach may be confusing in the context of Plasmodium, where mitosis is closed and neither astral microtubules nor a canonical central spindle have been demonstrated.

In the revised Figure 1b, we have therefore:

- (i) removed astral microtubules,
- (ii) depicted a closed nucleus with the MTOC embedded in the nuclear envelope, and
- (iii) avoided implying canonical midzone structures.

We chose to retain a generalised “cytokinesis-associated zone” to indicate that some Aurora paralogs, particularly ARK3 in *Toxoplasma*, are located at structures involved in karyokinesis/cytokinesis. This is meant to imply a functional analogy, not a homologous central spindle, and is now presented more cautiously in the revised panel and legend.

- I don't think it is fair to say ARK1-3 are absent in *Cryptosporidium* there are 6 *C. parvum* proteins in the ARK Orthogroup (Cdg1_1220, Cgd1_2630, Cgd8_750, Cgd3_3040, Cgd2_1830, Cgd6_5240). Comparing these proteins against the consensus sequences in Figure 6, it seems likely that Cgd3_3040 is the *C. parvum* homologue of ARK2.

We thank the reviewer for raising this point. In part the difficulty is that automated clustering approaches differ markedly in their ability to capture orthology for highly divergent kinase families—a situation also encountered with Bub1. In that case, early analyses concluded that Plasmodium lacked Bub1 because candidate kinases were extremely divergent and lacked the characteristic N-terminal TPR domain. However, as mentioned above, recent analyses have identified a highly divergent, Bub1 ortholog in Plasmodium. This underscores a general point: for rapidly evolving kinase families, inference of orthology requires explicit phylogenetic and structural analysis rather than reliance on clustering-based orthogroups alone. There is a similar

challenge for the Aurora/ARK kinases. Here, inference of orthology is complicated by multiple parallel gene family expansions and extensive lineage-specific divergence, particularly in apicomplexans. Because the Aurora family is much closer to other eukaryotic kinase families than is Bub1, incorrect grouping by automated tools is more likely.

This is precisely what occurs between OrthoDB version 6 and version 7:

OrthoDB v6 (correct reconstruction)

- OrthoDB v6 clusters Cgd2_3190 together with ARK1–2–3 from Plasmodium and Toxoplasma.
- This grouping reflects the true phylogenetic unity of the apicomplexan Aurora/ARK family.
- In other words, Cgd2_3190 is the only bona fide Aurora-related kinase in Cryptosporidium.

OrthoDB v7 (source of the confusion)

- OrthoDB v7 expanded clustering thresholds, which caused the apicomplexan Aurora paralogs to be split across multiple orthogroups.
- OG7_0000004 contains Cgd2_3190 together with ARK1 orthologues — this is the correct orthogroup.
- OG7_0000103, referenced by the reviewer, contains ARK2 and ARK3 mixed with unrelated kinase families.
 - o This heterogeneous composition indicates a clustering artefact, not true orthology.

Thus, the presence of multiple *C. parvum* kinases in OG7_0000103 does not imply that Cryptosporidium possesses ARK2 or ARK3 homologues. Rather, it reflects the same issue we saw with Bub1: when kinase families are highly divergent, automated orthogroup clustering is insufficient and must be complemented by targeted phylogenetic and structural analysis.

Our own analyses (based on phylogenetic reconstruction, active-site motif comparison, domain architecture) consistently show that:

- Cgd2_3190 is the sole Aurora/ARK-like kinase in Cryptosporidium,
- ARK2 and ARK3 are apicomplexan-specific expansions, restricted to hematozoans and coccidians. The Cryptosporidium IDs that are mentioned by the reviewer are different kinases altogether.

• Considering the strength and frequency of all the phenotypes quantified in Figure 3, it seems surprising that there is a <50% growth defect at this point in time. I think this is worth

discussing in more detail, and also mentioning whether the reasonably modest growth defect might be due to the compensatory upregulation of ARK2.

We agree with the reviewer that upregulation of ARK2 may compensate in part for the loss of PfARK1 and have mentioned this possibility in the Discussion (Lines 514-518).

- In Extended Data Figure 2d, the Hoechst staining seems to be almost fully excluded from the forming merozoites. Is there a channel alignment issue or something there?

We have replaced the image with a higher-quality version that clearly shows Hoechst staining in the forming merozoites (Extended Data Fig. 2d).

- The expression profiles of ARK1 and AMA1 are very different. Did the authors determine if the *ark1*-PTD parasites have approximately normal blood-stage growth? This is important for interpreting the results of the bite-back experiment. Also, cause AMA1 is expressed in sporozoites, would it be expected to get a second wave of ARK1 expression during late oocyst development?

We appreciate these insightful comments. During blood stage asexual growth, although the timing of expression of ARK1 and AMA1 differs slightly, the *amal* promoter is strongly active during schizogony. We monitored the asexual growth of *ark1*-PTD parasites and found it to be comparable to that of the WT control, confirming that ARK1 expression driven by the *amal* promoter is sufficient to support blood-stage replication. This implies that the delayed patency in the bite-back experiment is due to the low number of transmitted sporozoites rather than a slow parasite growth rate in the mouse.

We agree that given the expression profile of AMA1, ARK1 is likely to be expressed in sporozoites of the *ark1*-PTD line. However, since the primary defect occurs during male gametogony, there is a severe reduction in oocyst formation and the absolute number of sporozoites is very low. Thus, any potential expression of ARK1 at this late stage of development in the mosquito is insufficient to rescue the transmission phenotype.

- If I am interpreting Extended data Figure 5 correctly, roughly 50% of sporozoites produced sporozoites make it to the salivary gland in both WT and *ark1*-PTD parasites. From this, can it be inferred that the sporozoites that get produced are likely still invasive?

We agree that the absolute number of parasites is reduced, but the proportion of sporozoites successfully migrating to the salivary glands is comparable to that of the WT line. Thus, the few successfully differentiated *ark1*-PTD sporozoites are likely functional and invasive. This suggestion is supported by our bite-back experiments, since infection was established (albeit with a delay), confirming that the sporozoites are indeed invasive. We have added a sentence to the Results section to highlight the fact that the transmission defect is primarily driven by the reduction in sporozoite numbers rather than a loss of invasiveness (Lines 298-300).

We also have corrected Extended Data Fig. 6c, as we identified an error in the plotting of this graph. The revised figure now accurately reflects the data. This change does not affect our interpretation of the results.

- Figure 4e is missing some scale bars and the ark1PTD oocysts take up less of the FOV than the WT, making them look much smaller than the quantification would indicate.

We apologise for this oversight and agree that the size of the scale bars in the original Figure 4e (now Figure 5e) was incorrect. The appropriate scale bars are now included in Figure 4e (now Figure 5e).

- Figure 3f, DMSO Tubulin is hugely over-exposed

We have reprocessed the image and included it in the figure (revised Figure 4f).

- Figure 2h the Hoechst looks very strange in the 3D-SIM image, is this expected?

The unusual appearance of the Hoechst signal in the 3D-SIM image was due to the selection of the focal plane. In this image, the focus was optimized to resolve the PbARK1-GFP signal in the apical region of the nucleus. However, due to the shallow depth of field in 3D-SIM, the signal from the bulk of the nuclear DNA (the Hoechst signal), which lies beneath this focal plane, was not fully captured, resulting in a weaker signal. To prevent confusion, we have added a clarification to the Figure Legend (revised Figure 3i)

Spelling, grammar, syntax, etc.

- Lines 57 and 63: De novo is not italicised.

We revised the manuscript (Lines 61 and 68).

- Line 125 should say nuclei, or parasite nuclei, instead of parasites

We revised the manuscript (Line 137).

- Line 140: I think “along microtubules” rather than “at microtubules” is probably a better description of the localisation

We revised the manuscript (Line 152).

- Line 275: ‘bite-back experiments’ should be explained in more detail

We have revised the text to explain bite back experiments (Lines 291-298).

- Line 312-313: “led us to examine whether this” can be deleted

We revised the text (Line 319).

- Line 339-340: I don’t understand “to confirm the logic of the observed subcellular localisations”.

Our intent was to state that the subcellular location of ARK1 is expected to reflect its interactions with specific scaffolding or regulatory proteins, such as INCENP-A/-B. However, since this causal link was not explicit in the original text, so the phrasing was unclear. We have rephrased this sentence to articulate this assumption directly:

Lines 348-350: To understand whether ARK1 localisation results from recruitment to specific regulatory scaffolds, we investigated which interacting proteins might direct ARK1 activity and positioning during both asexual and sexual stages of the parasite life cycle.

- Line 353-358 is more discussion rather than results

We agree that the interpretation of the interactome data, specifically regarding the absence of Survivin/Borealin and its evolutionary implications, is better suited to the Discussion. Accordingly, we have moved this text into the Discussion, where we address the composition of the CPC (Lines 465-484).

- Extended data Figure 2c: What do the two annotated sizes correspond to?

The two annotated sizes correspond to full-length PbARK1-GFP fusion protein (~68 kDa) and the control, PK2-GFP (~80 kDa). We have repeated this western blot analysis, using WT-GFP parasites as a control. In the revised figure, the ~27 kDa band in the control lane corresponds to GFP and in the ARK1-GFP lane, the ~68 kDa upper band corresponds to the full-length ARK1-GFP protein, while the lower band likely represents truncated ARK1-GFP containing GFP sequence (Extended Data Fig. 2c).

References

1. Roques M, Niu C, Brochet M, Brusini L. A modular chromosomal passenger complex rewires chromosome segregation in *Plasmodium berghei*. *bioRxiv*, (2025).
2. Rawat RS, *et al.* Protein kinase PfPK2 mediated signalling is critical for host erythrocyte invasion by malaria parasite. *PLoS Pathog* **19**, e1011770 (2023).

3. Birnbaum J, *et al.* A genetic system to study *Plasmodium falciparum* protein function. *Nat Methods* **14**, 450-456 (2017).
4. Wyss M, Thommen BT, Kofler J, Carrington E, Brancucci NMB, Voss TS. The three *Plasmodium falciparum* Aurora-related kinases display distinct temporal and spatial associations with mitotic structures in asexual blood stage parasites and gametocytes. *mSphere* **9**, e0046524 (2024).
5. Reininger L, Wilkes JM, Bourgade H, Miranda-Saavedra D, Doerig C. An essential Aurora-related kinase transiently associates with spindle pole bodies during *Plasmodium falciparum* erythrocytic schizogony. *Mol Microbiol* **79**, 205-221 (2011).
6. Rawat A, *et al.* PfPPM2 signalling regulates asexual division and sexual conversion of human malaria parasite *Plasmodium falciparum*. *Nat Commun* **16**, 4790 (2025).
7. Bushell E, *et al.* Functional Profiling of a *Plasmodium* Genome Reveals an Abundance of Essential Genes. *Cell* **170**, 260-272 e268 (2017).
8. Solyakov L, *et al.* Global kinomic and phospho-proteomic analyses of the human malaria parasite *Plasmodium falciparum*. *Nat Commun* **2**, 565 (2011).
9. Tewari R, *et al.* The systematic functional analysis of *Plasmodium* protein kinases identifies essential regulators of mosquito transmission. *Cell Host Microbe* **8**, 377-387 (2010).

Response to reviewers

We thank the reviewers for their continued assessment of our work and their constructive feedback. We are pleased that Reviewers #1 and #3 find the manuscript ready for publication. We have addressed the remaining suggestions from Reviewer #2 regarding figure presentation and the statistical correction pointed out by Reviewer #3. We also have corrected Figure 5a and 7a as we noticed an older version of these panels was inadvertently used.

Reviewer comments

Reviewer #1 (Remarks to the Author):

The authors have adequately addressed my previous concerns. This is a high-quality study that provides new insights into the MTOC of Plasmodium parasites.

We thank the reviewer for the positive evaluation and the time in reviewing our manuscript.

Reviewer #2 (Remarks to the Author):

The authors have addressed most comments in this revised version, and the clarity of the manuscript has improved. To further strengthen data presentation, as noted in my previous major comments 2, 3 and 5, I encourage the inclusion of zoomed-in views of individual mitotic spindle structures in Fig. 2b and zoomed-in views of the highlighted phenotypic regions in Fig. 4d and 4f of *P. falciparum*, similar to Fig. 5g for *P. berghei*. Without such magnification, several described phenotypes-such as “centrin aggregates were seen away from the nuclei and disorganised” (line 229) in Fig. 4d and “subpellicular MT were either not observed or were mislocalised” (lines 249-250) in Fig. 4f- are difficult to discern.

We have included zoomed-in views in Figures 2b, 4d, 4f, as recommended and thank the reviewer for the suggestion. In order to accommodate extra panels, the indicated figures are slightly re-arranged.

I was also unable to locate Extended Data Fig. 3d-e referenced in the response to minor comment 5.

In this response, we intended to refer to Extended Data Fig. 2d-e (Supplementary Fig. 2d-e). We apologise for the error.

Reviewer #3 (Remarks to the Author):

The authors have done a very thorough job responding to all of my comments, and these changes have no impact on the significance/novelty of the findings of this study. In my opinion this study is now ready for publication.

My only unaddressed comment is regarding the statistical analysis in Figure 4a. Here the authors use ANOVA to calculate what they're presenting as a pairwise analysis (which should be a t-test or similar instead). In either case, statistical tests where the algorithm includes sample variance (like ANOVA or t-tests) cannot be performed on this data because the variance of the DMSO treated sample = 0 due to normalisation. This means the statistical significance of these comparison gets inflated because true sample variances are not being compared.

We have redone the inset graphs and the related stats in Figure 4a and Extended Fig. 5a, b and performed pairwise t-test for each individual time point (48, 96 and 144 h) to compare the parasitemia between DMSO and RAP-treated parasites at these time points.